# Dissecting the role of the human microbiome in COVID-19 via metagenome-assembled genomes

Shanlin Ke [1], Scott T. Weiss[1] & Yang-Yu Liu [1]✉

Coronavirus disease 2019 (COVID-19), primarily a respiratory disease caused by infection with Severe Acute Respiratory Syndrome Coronavirus 2 (SARS-CoV-2), is often accompanied by gastrointestinal symptoms. However, little is known about the relation between the human microbiome and COVID-19, largely due to the fact that most previous studies fail to provide high taxonomic resolution to identify microbes that likely interact with SARS-CoV-2 infection. Here we used whole-metagenome shotgun sequencing data together with assembly and binning strategies to reconstruct metagenome-assembled genomes (MAGs) from 514 COVID-19 related nasopharyngeal and fecal samples in six independent cohorts. We reconstructed a total of 11,584 medium-and high-quality microbial MAGs and obtained 5403 non-redundant MAGs (nrMAGs) with strain-level resolution. We found that there is a significant reduction of strain richness for many species in the gut microbiome of COVID-19 patients. The gut microbiome signatures can accurately distinguish COVID-19 cases from healthy controls and predict the progression of COVID-19. Moreover, we identified a set of nrMAGs with a putative causal role in the clinical manifestations of COVID-19 and revealed their functional pathways that potentially interact with SARS-CoV-2 infection. Finally, we demonstrated that the main findings of our study can be largely validated in three independent cohorts. The presented results highlight the importance of incorporating the human gut microbiome in our understanding of SARS-CoV-2 infection and disease progression.

The ongoing pandemic of coronavirus disease 2019 (COVID-19), a respiratory disease caused by severe acute respiratory syndrome coronavirus 2 (SARS-CoV-2), has infected billions of people worldwide. A broad range of clinical manifestations of COVID-19 has been reported, including asymptomatic or mild disease with cough and fever to severe pneumonia with multiple organ failure and acute respiratory distress syndrome (ARDS) leading to death[1]. Existing studies found that a large proportion of COVID-19 patients had at least one gastrointestinal (GI) symptom[2–5], such as diarrhea, vomiting, or belly pain. Moreover, it has been reported that, among 73 SARS-CoV-2-infected hospitalized patients in China, 53.4% of patients tested positive for SARS-CoV-2 in their stool samples ranging from day 1 to 12 post infection[6]. Importantly, in more than 20% of infected patients, their fecal samples remained positive for the virus even after the respiratory and/or sputum samples exhibited no detectable virus[6]. In some cases, the viral load in feces is even higher than that in pharyngeal swabs[3]. All these results suggest that the GI tract might be an important extra-pulmonary site for SARS-CoV-2 infection. Currently, the role of angiotensin-converting enzyme 2 (ACE2) in the invasion of host cells by SARS-CoV-2 via its spike protein is well-established[7], and ACE2 is also highly expressed in the small intestine and colon[4,8]. Therefore, the

[1]Channing Division of Network Medicine, Department of Medicine, Brigham and Women's Hospital and Harvard Medical School, Boston, MA 02115, USA. ✉e-mail: yyl@channing.harvard.edu

prolonged presence of large amounts of fecal SARS-CoV-2 RNA virus is unlikely to be explained by the swallowing of virus particles replicated in the throat but rather suggests enteric infection with SARS-CoV-2.

The human GI tract is the largest immune organ in the body and plays a critical role in the immune response to pathogenic infection or commensal intrusion[9]. Trillions of microbes live inside the GI tract. Those microbes and their genes, collectively known as the human gut microbiome, modulate host immunity[10]. To date, several studies, based on 16 S rRNA gene sequencing, have demonstrated that the human upper respiratory and gut microbiome are broadly altered in patients with COVID-19[11–16]. Although 16 S rRNA gene sequencing provides valuable insights into the general characteristics of the human microbiota, it does not offer the taxonomic resolution needed to capture sufficient sequence variation to discriminate between closely related taxa[17]. Other studies, based on whole-metagenome shotgun (WMS) sequencing, explored the links between the human microbiome and SARS-CoV-2 infection by mapping short metagenomic reads to reference genome databases[18–23]. Despite the fact that the analysis of WMS sequencing data provides more information than 16 S rRNA gene sequencing data analysis, existing studies based on reference genome databases are subject to the limitations and biases of those databases and unable to characterize microbes that do not have closely related culture representatives. In fact, an estimated 40–50% of human gut species lack a reference genome[24,25]. This may result in a strong null bias in characterizing the gut microbial community.

An alternative strategy to analyze WMS sequencing data is to reconstruct metagenome-assembled genomes (MAGs) through de novo assembly and binning[26]. One key advantage of this strategy is that it allows recovery of genomes for microorganisms that have yet to be isolated and cultured and hence are absent from the current reference genome databases. This strategy has been adopted in several studies to provide genomic insights into microbial populations that are critical to human health and disease[27,28]. In this study, we applied state-of-the-art metagenome assembly and binning strategies to reconstruct microbial population genomes directly from microbiome samples of COVID-19 patients and controls (Fig. 1). Our major goals were to construct a COVID-19 related metagenomic genome catalog to identify novel taxa and strain-level differences that are likely related to the clinical manifestations of SARS-COV-2 infection. Our results demonstrate the association of the human microbiome and SARS-COV-2 infection at an unprecedentedly high level of taxonomic resolution. More importantly, our study provides a unique resource to directly investigate the genomic content of COVID-19 relevant microbial strains and sheds light on more targeted follow-up studies.

## Results

### COVID-19 related metagenomic datasets

To examine the relation between the human microbiome and COVID-19 via MAGs, we first gathered WMS sequencing data from the COVID-19 related human microbiome studies (publicly available as of August 2021) as the discovery cohorts. We collected the raw WMS sequencing data of 514 microbiome samples ($n = 359$ individuals) from six publicly available datasets (Fig. 1a and Table 1) with different technical settings (e.g., sequencing platform and sequencing depth), including nasopharyngeal ($n = 96$) and fecal microbiome ($n = 418$) samples. Among these samples in the discovery cohorts, we have 404 (78.60%) and 110 (21.40%) samples from COVID-19 patients and Non-COVID-19 controls (Fig. 2a), respectively.

To validate our key findings in the discovery cohorts, we collected the raw WMS sequencing data of 341 fecal microbiome samples ($n = 278$ individuals) from three publicly available datasets (Table S1, publicly available as of April 2022). In the validation cohorts, 62.46% and 37.54% microbiome samples from COVID-19 patients and Non-COVID-19 controls, respectively.

### A high-quality microbial genome catalog of COVID-19 constructed from the discovery cohorts

After quality control, we performed metagenomic assembly and binning on those microbiome samples from the discovery cohorts and recovered 12,195 MAGs in total (Fig. 1b). To standardize the genome quality across all datasets, we used thresholds of ≥50% genome completeness and ≤5% contamination[29,30], resulting in 11,584 MAGs [mean completeness = 87.55%; mean contamination = 0.99%; mean genome size = 2.6 megabases (Mb); mean N50 = 61.8 kilobases (kb), Figs. S1, S2]. Here N50 is the sequence length of the shortest contig at 50% of the total genome length. To obtain the view of the microbial community at the species level, we first organized 11,584 MAGs into species-level genome bins (SGBs) at an ANI (average nucleotide identity) threshold of 95%, resulting in a total of 872 SGBs, of which 160 (18.35%) SGBs represented species without any available genomes from the Genome Taxonomy Database (GTDB)[31] and were defined as unknown SGBs (uSGB, Fig. S3). To evaluate the highest quality representative genomes, we dereplicated the 11,584 MAGs at an ANI threshold of 99%, resulting in a final set of 5403 non-redundant MAGs (nrMAGs) with strain-level resolution [mean completeness = 86.87%; mean contamination = 0.99%; mean genome size = 2.4 megabases (Mb); mean N50 = 63.2 kilobases (kb), Fig. 2d, e and Fig. S2]. We found that each Non-COVID-19 microbiome sample contributed relatively higher rates of total MAGs and nrMAGs than COVID-19 microbiome samples as 21.40% Non-COVID-19 microbiome samples contributed to 31.32% of total MAGs and 38.94% of nrMAGs (Fig. 2a–c and Fig. S4).

Among those 5403 strain-level nrMAGs, 2,190 (40.53%) nrMAGs satisfied the medium-quality criteria (50% ≤ completeness < 90% and ≤5% contamination), and 3,213 (59.47%) nrMAGs showed high-quality (≥90% completeness and ≤5% contamination) (Fig. 1)[29,30]. Using the Genome Taxonomy Database[31], 5,397 (99.89%) and 6 (0.11%) nrMAGs were assigned to bacterial and archaeal domains, respectively (Fig. S5). The phyla information of nrMAGs was summarized in Fig. 2d.

### Alterations of the human microbiome in COVID-19 patients

Previous studies demonstrated that SARS-CoV-2 infection is associated with the alpha diversity of the human gut[12,32,33] and oral[34,35] microbiome at the genus- or species-level. We first investigated in our discovery cohorts whether SARS-CoV-2 infection is associated with alpha diversity of the human microbiome at the nrMAG-level. The alpha diversity measures (i.e., Richness and Shannon index) from COVID-19 patients and Non-COVID-19 controls were compared (Fig. 3a and Fig. S6). In accordance with previous studies[11,32], we found that the Richness and Shannon index of the gut microbiome in COVID-19 patients were significantly lower than that in Non-COVID-19 controls in two datasets (Zuo et al.[18] and Yeoh et al.[19], Fig. 3a and Fig. S6a, d). Interestingly, no significant results were found between patients with COVID-19 and Non-COVID-19 Pneumonia controls (Fig. 3a and Fig. S6a). Consistent with a previous study[36], we found no significant differences between COVID-19 patients and Non-COVID-19 controls in the nasopharyngeal microbiome samples (Fig. 3a and Fig. S6e, f). This may be due to the small sample size and the fact that Non-COVID-19 controls are not health controls in one of the nasopharyngeal microbiome datasets (Liu et al.[36]). Moreover, in the other nasopharyngeal microbiome dataset (PRJNA743981[37]), we only identified a small number of nrMAGs, as a large portion of sequencing reads from this dataset were contamination from the human genome. In accordance with the two discovery cohorts (i.e., Zuo et al.[18] and Yeoh et al.[19]), we found that the alpha diversity of the gut microbiome in COVID-19 patients was overall lower than that in Non-COVID-19 controls in the validation cohorts (Fig. S7).

In line with previous studies[11,38], PCoA (principal coordinates analysis) combined with PERMANOVA (permutational multivariate

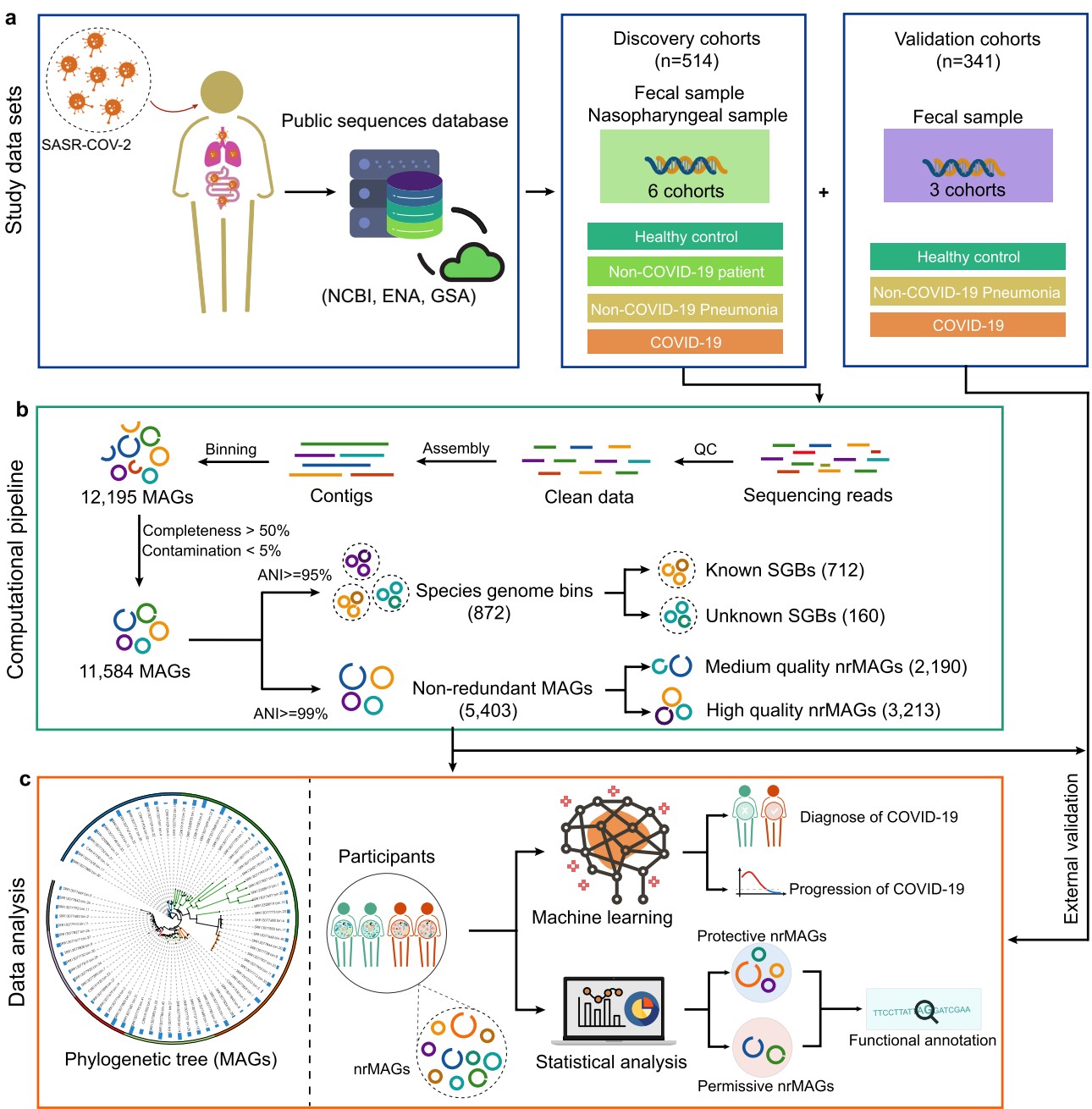

**Fig. 1 | Conceptual framework of study. a** To understand the relation between human microbiome and COVID-19 via metagenome-assembled genomes (MAGs), we collected a total of 514 (6 cohorts) and 341 (3 cohorts) shotgun metagenomic sequencing data on the discovery and validation cohorts, respectively. These microbiome sample including fecal and nasopharyngeal samples from COVID-19 and non-COVID-19 controls. **b** A total of 11,584 MAGs (≥50% completeness and ≤5% contamination) were constructed from metagenomic sequencing data from the discovery cohorts. The reconstructed MAGs were first clustered to 872 species-level genome bins (SGBs) at 95% of the ANI (average nucleotide identity). SGBs containing at least one reference genome (or metagenome-assembled genome) in the Genome Taxonomy Database (GTDB) were considered as *known* SGBs. Otherwise, they were considered as *unknown* SGBs. The reconstructed MAGs were then dereplicated to 5403 non-redundant MAGs (nrMAGs, strain level) based on 99% of ANI. The 5403 nrMAGs were divided into medium-quality MAGs (50% ≤ completeness < 90% and ≤5% contamination) and high-quality MAGs (≥90% completeness and ≤5% contamination). **c** The phylogenetic tree of nrMAGs was constructed using PhyloPhlAn. We employed Random Forest machine learning models together with nrMAGs to diagnose COVID-19 and predict the progression of COVID-19 (date of negative RT-qPCR results). The permissive and protective nrMAGs of COVID-19 were identified by GMPT pipeline. And the genomes of permissive and protective nrMAGs were functionally annotated using Prokka and MicrobeAnnotator. The main findings from the discovery cohorts were then validated using the data from three independent cohorts.

analysis of variance) revealed that the two discovery datasets (from Zuo et al.[18] and Yeoh et al.[19]) had a significant difference in the gut microbial community structure between the patients with COVID-19 and Non-COVID-19 controls at the nrMAG-level (Fig. 3c, d and Fig. S8). Similar results were also found in the validation cohorts (Fig. S9a-c). Moreover, the patients with COVID-19 in these two discovery datasets showed significant higher within-group variation than that in the Non-COVID-19 controls (Fig. 3b). And this pattern was confirmed in two validation cohorts (Xu et al.[39] and Li et al.[22], Fig. S9e, f).

In the dataset from Yeoh et al.[19], the gut microbiome samples from those patients with COVID-19 were collected before and after their nasopharyngeal aspirates or swabs tested negative for SARS-CoV-2 via

**Table 1 | The discovery cohorts of human metagenome datasets analyzed in this study**

| Dataset | Zuo et al.[18] | Britton et al.[53] | Cao et al.[54] | Yeoh et al.[19] | Liu et al.[36] | PRJNA743981[37] | Total |
|---|---|---|---|---|---|---|---|
| COVID-19 positive samples | 50 | 36 | 37 | 196 | 6 | 79 | 404 |
| COVID-19 negative samples | 21 | 0 | 0 | 78 | 3 | 8 | 110 |
| Total subjects | 36 | 36 | 13 | 178 | 9 | 87 | 359 |
| Total samples | 71 | 36 | 37 | 274 | 9 | 87 | 514 |
| Longitudinal | Yes | No | Yes | Yes | No | No | – |
| Source | Fecal | Fecal | Fecal | Fecal | Nasopharyngeal | Nasopharyngeal | – |
| Geography | CHINA | USA | CHINA | CHINA | CHINA | – | – |
| Year | 2020 | 2021 | 2021 | 2021 | 2021 | 2021 | – |
| Sequencing platform | Illumina Novaseq 6000 | Illumina Hiseq | HiSeq XTen (PE250) or NovaSeq (PE150) | Illumina Novaseq 6000 | Illumina Novaseq 6000 | Illumina Novaseq 6000 | – |
| Total sequences (mean) | 17,245,724.18 | 4,636,385.22 | 19,666,057.43 | 23,717,063.99 | 40,593,115.44 | 28,683,197.33 | – |
| Accession number | PRJNA624223 | PRJNA660883 | PRJCA003532 | PRJNA650244 | PRJNA656660 | PRJNA743981 | – |

#COVID-19 negative samples are healthy controls or Non-COVID-19 patients who tested negative for SARS-CoV-2 infection.

RT-qPCR. Furthermore, patients with COVID-19 were classified into four severity groups (i.e., mild, moderate, severe, and critical) based on symptoms as reported in the previous study[40]. We found that patients with COVID-19 who had milder disease severity showed significant higher Shannon diversity in their gut microbiome (Fig. 3e). Interestingly, the composition of nrMAGs in patients with COVID-19 after recovery (negative for SARS-CoV-2 via RT-qPCR) were significantly different from Non-COVID-19 controls than from patients with COVID-19 before recovery (positive for SARS-CoV-2 via RT-qPCR, Fig. 3f). In line with a previous study at the metabolic capacity level[41], these results indicate that the gut microbiome of patients with COVID-19 did not return to a relatively healthy status right after their recovery from SARS-CoV-2 infection. We then observed that disease severity of COVID-19 was significantly positively associated with the gut microbiome dissimilarity between COVID-19 patients and Non-COVID-19 controls (Fig. 3g). To understand the relation between disease severity and short-term variation in the gut microbiome of patients with COVID-19, we traced the changes in the microbiome within each individual associated with disease severity. Interestingly, patients with COVID-19 who had milder disease severity showed lower temporal variation in the gut microbiome (quantified by the Bray-Curtis dissimilarity of longitudinal microbiome samples, Fig. 3h). The lower temporal variation of gut microbiome samples in milder disease severity groups is partly due to the higher fraction of common nrMAGs across the longitudinal microbiome samples in those groups (Fig. 3i).

**COVID-19 patients lost many strains of multiple microbial species**
To explore whether the strain-level diversity within the same species is related to COVID-19, we analyzed data from the two discovery cohorts (Zuo et al.[18] and Yeoh et al.[19]) as well as the three validation cohorts (Zhang et al.[41], Xu et al.[39], and Li et al.[22]). We first grouped all the nrMAGs to the species level based on GTDB taxonomy information. For each species, we computed its strain richness (i.e., the number of its nrMAGs) for all microbiome samples. Those nrMAGs without the species annotation and species containing only one nrMAGs were excluded. Interestingly, the top-30 microbial species with the highest strain richness were highly overlapped between the discovery and validation cohorts (Fig. 4a, b, Figs. S10a-c, S11). Notably, we found that COVID-19 patients lost many strains of many microbial species when compared to Non-COVID-19 controls for both the discovery (Fig. 4c, d) and the validation cohorts (Fig. S10d-f). Moreover, those species with significant strain loss are highly overlapped between discovery cohorts (Fig. S12a), including *Bariatricus comes*, *Blautia_A obeum*, *Blautia_A*

*wexlerae*, *Dorea formicigenerans*, *Faecalibacterium prausnitzii_D*, *Faecalibacterium sp900539945*, and *Fusicatenibacter saccharivorans*. Importantly, these results are highly consistent with the alpha diversity analysis at the nrMAG-level that COVID-19 patients had significantly lower number of nrMAGs identified than that of Non-COVID-19 controls on discovery cohorts (i.e., Zuo et al. and Yeoh et al.). We found that some of microbial species (9 of 30) with high COVID-19 related strain loss in the discovery cohorts were also identified in the validation cohorts (Fig. S12c). For the Xu et al.[39] cohort, several species from COVID-19 patients showed significantly higher number of strains than that of Non-COVID-19 controls. This may be due to the fact that microbiome samples of COVID-19 patients (collected in 2020) and Non-COVID-19 controls (collected in 2016) were not collected and sequenced at the same time.

We next investigated the disease severity in relation to the strain diversity using data from Yeoh et al.[19]. We found seven species whose strain richness were positively associated with disease severity (Spearman correlation coefficients ≥ 0.9, Supplementary Data 1), including *Enterocloster bolteae*, *Fournierella sp900543285*, *Hungatella effluvii*, *Lacticaseibacillus rhamnosus*, *Ligilactobacillus ruminis*, and *Ligilactobacillus salivarius*. Moreover, a total of 222 microbial species' strain richness were negatively correlated with disease severity (Spearman correlation coefficients ≤ −0.9, Supplementary Data 1), such as *Blautia_A obeum*, *Bariatricus comes*, *Blautia_A wexlerae*, and *Faecalibacterium prausnitzii_D*.

**nrMAGs accurately classifies COVID-19 patients and Non-COVID-19 controls**
Previous studies have demonstrated the diagnostic potential of the microbiome-based classification for SARS-CoV-2 infection using genus- or species-level taxonomic profiles[12,22,42]. To test whether the gut microbial composition at the nrMAG-level can distinguish COVID-19 patients from Non-COVID-19 controls, we built random forest classifiers on two datasets (Zuo et al.[18]: 50 patients with COVID-19 and 15 Non-COVID-19 controls; and Yeoh et al.[19]: 196 patients with COVID-19 and 78 Non-COVID-19 controls), separately. Importantly, this analysis was performed with 5-fold cross-validation and the data were randomly split into training and test sets 50 times. Since we had unbalanced classes, we applied two metrics to quantify the classification performance: AUROC (Area Under the Receiver Operating Characteristic curve) and AUPRC (Area Under the Precision-Recall Curve). Consistent with the PCoA analysis (Fig. 3c), using the data from Zuo et al.[18], we found that nrMAGs can accurately detect COVID-19 with the mean AUROC and AUPRC values of 0.981 and 0.971, respectively (Fig. S13a). The top COVID-19 related features included

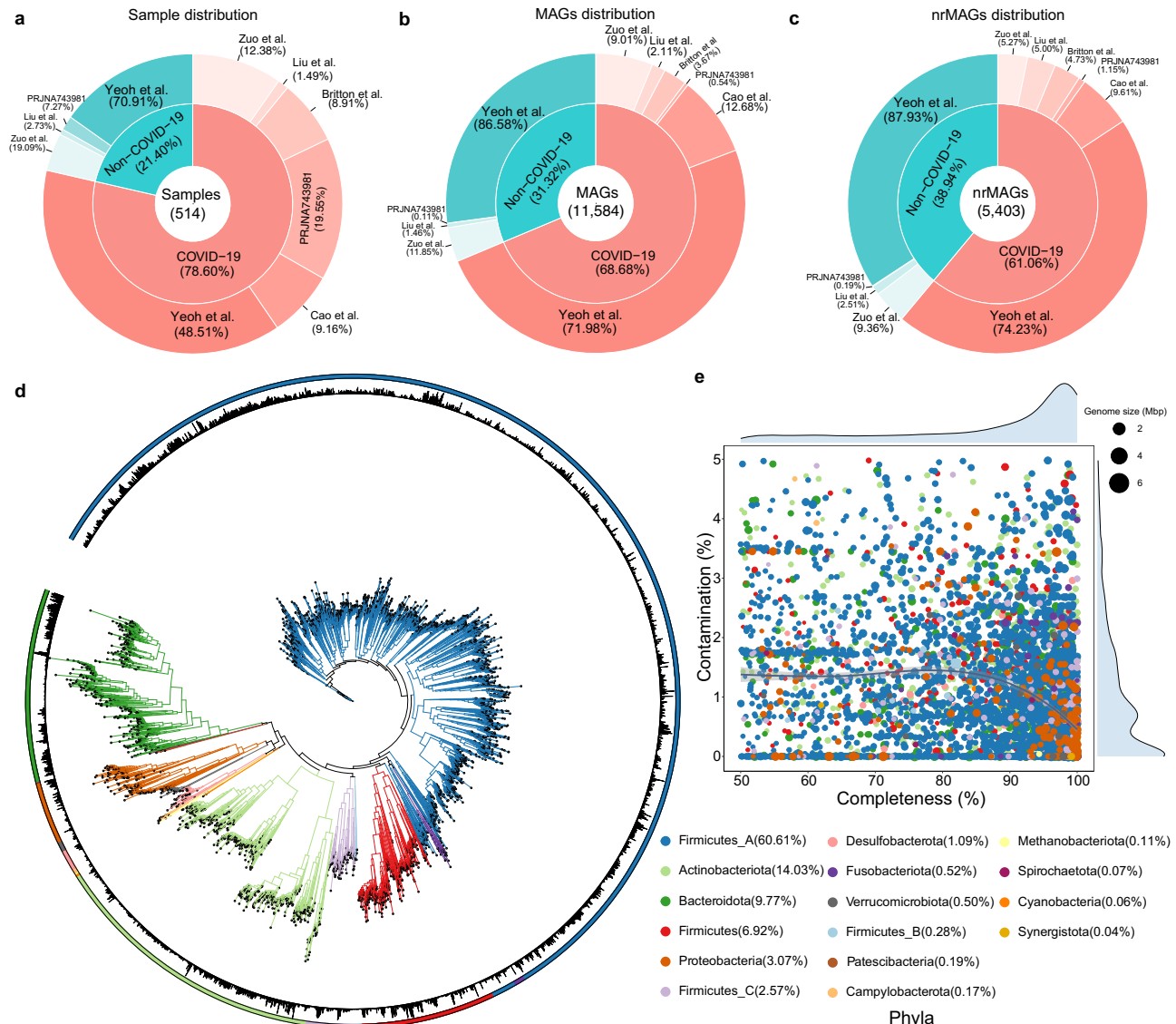

**Fig. 2 | Reconstruction of MAGs from 514 COVID-19 related shotgun metagenomics sequencing data in the discovery cohorts. a** Sample distribution among different dataset and disease status. **b** Number of MAGs recovered from different dataset and disease status. **c** Number of nrMAGs recovered from different dataset and disease status. **d** Phylogenetic tree of nrMAGs constructed using PhyloPhlAn. The color of outer cycle and clades represents phylum and bar plot within cycle represents the average relative abundance across all microbiome samples. **e** The distribution of completeness and contamination on nrMAGs and the color of point represents phylum. And the size of point represents the genome size of nrMAGs.

multiple nrMAGs from *Blautia_A sp003480185, Blautia_A wexlerae, Agathobacter faecis, Eisenbergiella sp900066775, Faecalibacterium prausnitzii_G,* and *Lachnospira rogosae* (Fig. S13b). Consistent with the first dataset (Zuo et al.[18]) and the PCoA analysis (Fig. 3d), the random forest classifier on the larger cohort (Yeoh et al.[19]) also showed high classification performance (AUROC ~ 0.920; AUPRC ~ 0.884; Fig. S13c). The key discriminatory nrMAGs of COVID-19 in this cohort belonged to *Adlercreutzia equolifaciens, Blautia_A sp003471165, Eisenbergiella sp900066775, Eubacterium I, Gemmiger sp900539695, and Romboutsia timonensis* (Fig. S13d). Moreover, three specific nrMAGs were identified (from *Mediterraneibacter_A butyricigenes* and *Eisenbergiella sp900066775)* as common features between the two datasets.

To further evaluate the generality of COVID-19 microbiome features in machine learning models, we first performed cross-validation between the two discovery cohorts of Zuo et al.[18] and Yeoh et al.[19]. Briefly, we trained our machine learning model with samples from the cohort of Zuo et al. (or Yeoh et al.) and then test the model with

samples from the cohort of Yeoh et al. (or Zuo et al.), respectively. The machine learning model trained with the data of Zuo et al. achieved an overall classification performance of AUROC ~ 0.798 and AUPRC ~ 0.607 when tested with samples from Yeoh et al. (Fig. S14a). Notably, we found the classifications trained with samples from Yeoh et al. can almost perfectly distinguish COVID-19 patients from Non-CONID-19 controls in the study of Zuo et al. (AUROC ~ 0.9994; AUPRC ~ 0.9991, Fig. S14b). To better understand those results, we also outputted the top-30 most important features ranked based on the mean decrease accuracy (MDA). Here, the MDA of a feature means its average accuracy loss after excluding this feature from the model. We found that the most important nrMAGs identified from the study of Zuo et al. have quite distinct abundance distributions between cases and controls in the study of Zuo et al. but not in the study of Yeoh et al (Fig. S15a, b). This explains the lower performance of the model (trained with data from Zuo et al.) in testing data from Yeoh et al. Moreover, the most important nrMAGs identified from the study of Yeoh et al. showed distinct abundance distributions between cases and

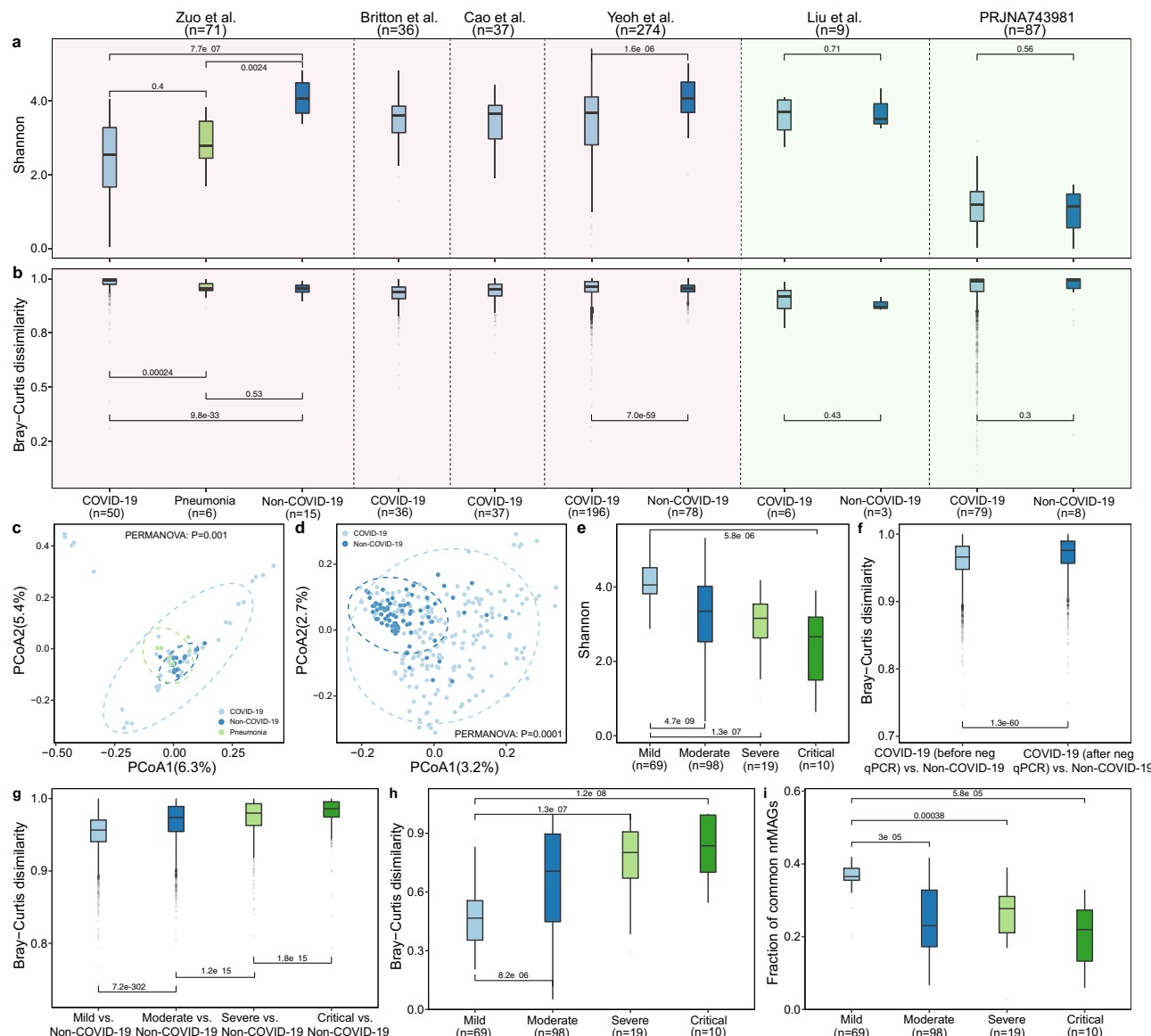

**Fig. 3 | COVID-19 related alterations of the human microbiome in the discovery cohorts.** Shannon diversity (**a**) and within group Bray−Curtis dissimilarity (**b**) of the human microbiome at the nrMAG-level from each study. The background color of each panel (**a**, **b**) represents the source of the microbiome samples from the human gut (light red) or nasopharynx (light green). Principal Coordinates Analysis (PCoA) plot based on Bray−Curtis dissimilarity of microbial compositions from the study of Zuo et al. (**c**) and Yeoh et al. (**d**). All PERMANOVA tests were performed with 9999 permutations based on Bray−Curtis dissimilarity, two-sided. **e** Shannon diversity at different disease severity groups from the study of Yeoh et al. **f** Boxplot of the gut microbiome Bray−Curtis dissimilarity between healthy controls and patients with COVID-19 before or after their nasopharyngeal aspirates or swabs tested negative

for SARS-CoV-2 via RT-qPCR from the study of Yeoh et al. **g** Boxplot of the gut microbiome Bray−Curtis dissimilarity between healthy controls and patients with COVID-19 from different disease severity groups from the study of Yeoh et al. **h** Boxplot of Bray−Curtis dissimilarity of individual microbiome temporal changes over time from different disease severity groups from the study of Yeoh et al. **i** The fraction of common nrMAGs on patients with COVID-19 over time from the study of Yeoh et al. *P*-values were calculated by two-sided Wilcoxon−Mann−Whitney test. Boxplots with medians are shown; the lower and upper hinges correspond to the first and third quartiles (the 25th and 75th percentiles); the upper and lower whiskers extend from the hinge to the largest and smallest value no further than 1.5* interquartile range from the hinge; outliers are plotted by translucent circles.

controls in the study Zuo et al. (Fig. S15c, d). This explains the almost perfect performance of the model (trained with data from Yeoh et al.) in testing data from Zuo et al.

We further tested the generalization of COVID-19 related microbiome features on the three validation cohorts (i.e., Zhang et al.[41], Xu et al.[39], and Li et al.[22]). We found that the classification models trained with the data of Yeoh et al.[19] achieved an overall reasonable classification performance on those validation cohorts (Fig. S16).

## nrMAGs accurately predict the progression of COVID-19

We next investigated the association between nrMAGs and the progression of COVID-19. To explore this association, we employed

a random forest regression model to predict the date of negative RT-qPCR result using the data from Yeoh et al.[19] (with 196 microbiome samples from 100 COVID-19 patients, Fig. S17). The regression tasks were performed with 5-fold cross-validation and we then randomly split the data 50 times. Remarkably, this approach demonstrated that the dates of negative RT-qPCR result were well predicted by nrMAGs (Pearson correlation 0.425, *P*-value = 1e − 45, Fig. 5a). Among the top-30 (based on the percentage increase in mean squared error) most important nrMAGs (Fig. 5b), we identified multiple species such as *Citrobacter freundii*, *Enterocloster sp900543885*, *Citrobacter portucalensis*, *Parabacteroides distasonis* and *Veillonella parvula*. Notably, we also found some nrMAGs

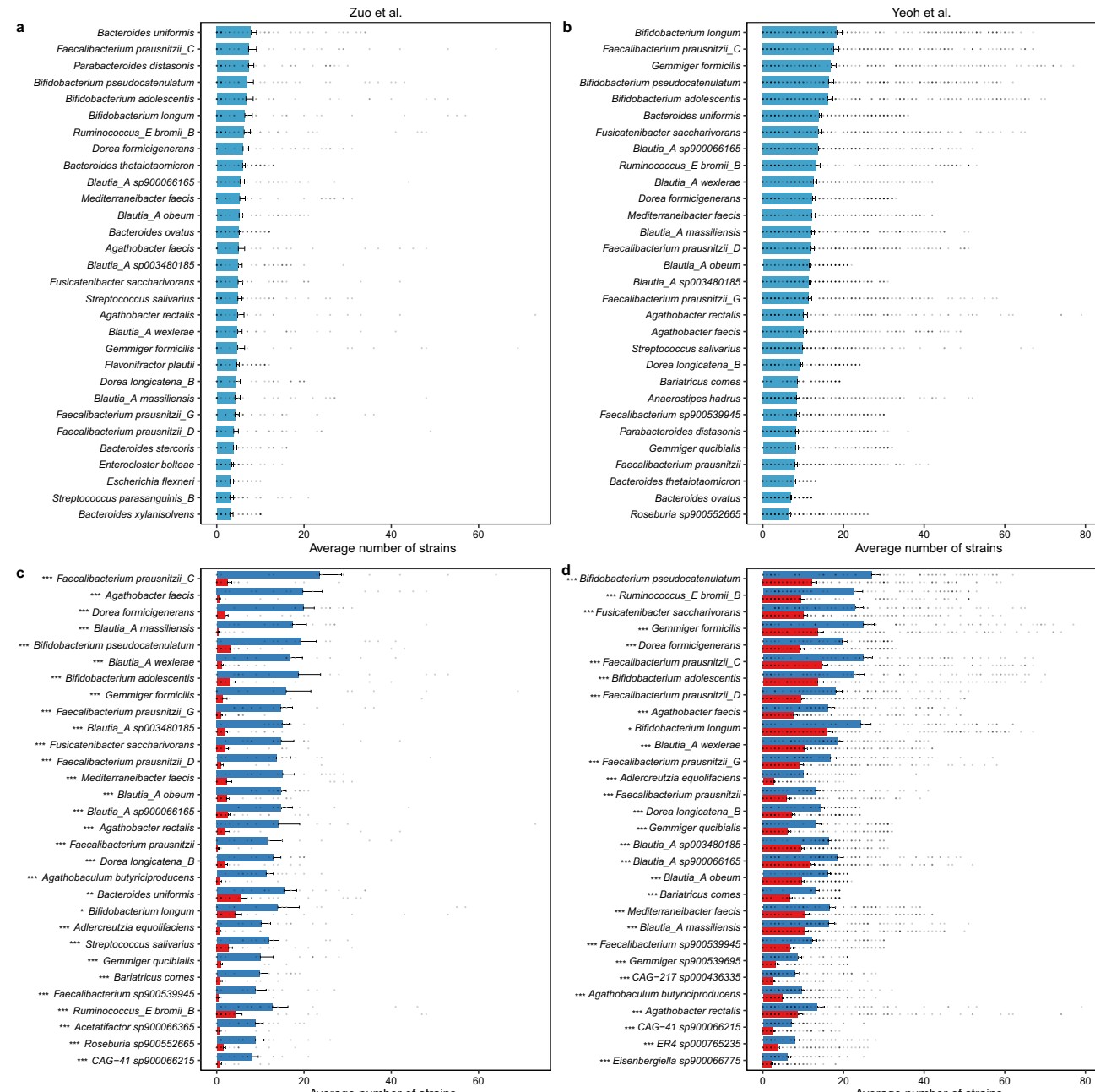

**Fig. 4 | COVID-19 related changes in strain richness of microbial species in the two discovery cohorts. a, b** The top-30 species with the highest strain-richness (i.e., number of nrMAGs) identified from the study of Zuo et al. (**a** sample size $n = 65$) and Yeoh et al. (**b** sample size $n = 274$). **c, d** The top-30 species with the highest strain-richness change between the Non-COVID-19 (blue) and COVID-19 (red) samples identified from the study of Zuo et al. (**c**) and Yeoh et al. (**d**). Data are presented as mean ± standard error of mean. *P*-values were calculated by two-sided Wilcoxon–Mann–Whitney test (ns nonsignificant, *$P < 0.05$; **$P < 0.01$, ***$P < 0.001$).

from well-known opportunistic pathogens including MAG02074 (*Klebsiella quasivariicola*[43]), MAG03769 (*Klebsiella pneumoniae*[44]), and MAG02080 (*Escherichia coli_D*[45]).

### Identification of putative permissive and protective nrMAGs for COVID-19 severity

To further characterize the relation between the human gut microbiome and COVID-19, we applied the generalized microbe-phenotype triangulation (GMPT) method to move beyond the standard association analysis[46] (Fig. 6a). Due to the availability of disease severity data, we first categorized participants from the study of Yeoh et al.[19] into five different disease severity groups (i.e., Non-COVID-19 healthy controls, mild, moderate, severe, and critical). The differentially abundant

nrMAGs were then calculated using ANCOM[47] (with each patient's identifier adjusted as a random effect) in the ten pair-wise comparisons. Using this approach, all pairwise differential abundance analyses yielded a total of 644 differentially abundant nrMAGs present in at least two pairwise comparisons. To understand the potential relationship between those candidate nrMAGs and COVID-19, we then calculated the Spearman correlation coefficients between the average relative abundances of nrMAGs and COVID-19 severity score (e.g., Non-COVID-19 healthy controls: 0; mild: 1; moderate: 2; severe: 3; and critical: 4) in different phenotype groups. Those differentially abundant nrMAGs with positive (or negative) Spearman correlation coefficients are potential permissive (protective) nrMAGs of COVID-19. Based on the frequency (≥6) of all pairwise comparisons ($n = 10$), we

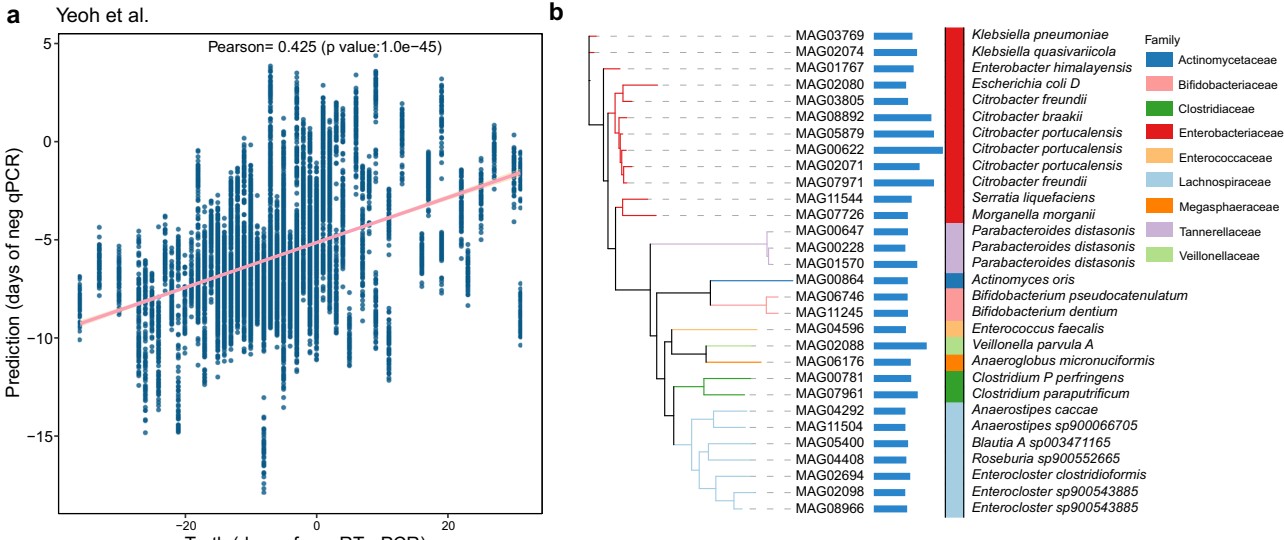

**Fig. 5 | The nrMAG-based machine learning model predicts the progression of COVID-19. a** Pearson correlation coefficient between the true and predicted date of negative RT-qPCR result on the random forest regression models. **b** The top-30 important nrMAGs related to performance of prediction. The importance of each feature in the regression was quantified by percent increase in mean square error (% IncMSE). The length of horizontal bar represents the mean %IncMSE value. The colors of vertical bar represent the taxonomy information of nrMAGs at the family level. The phylogenetic tree of these nrMAGs was constructed using PhyloPhlAn.

summarized the results from GMPT in Fig. 6b and Supplementary Data 2. This analysis identified a total of 74 nrMAGs that were associated with SARS-CoV-2 infection, including 8 permissive (Spearman correlation > 0) nrMAGs, 63 protective (Spearman correlation < 0) nrMAGs, and 3 neutral nrMAGs (Spearman correlation = 0).

We identified multiple species with highly similar genomes from those permissive nrMAGs (Fig. 6b), including *Enterocloster bolteae* (3 nrMAGs), *Anaeroglobus micronuciformis*, *Hungatella effluvii* (3 nrMAGs), and *Enterococcus_B faecium*. Consistent with previous reports that the gut microbiome of COVID-19 patients showed significant higher abundance of *Enterococcus faecium* compared to health controls[48]. Moreover, we found that the strain richness (number of nrMAGs) of two permissive species (i.e., *Enterocloster bolteae* and *Hungatella effluvii*) positively associated with disease severity (Supplementary Data 1). Among the 63 protective nrMAGs, the dominant species were *Blautia_A obeum* (13 nrMAGs), *Bariatricus comes* (9 nrMAGs), *Faecalibacterium prausnitzii_D* (6 nrMAGs), *Blautia_A wexlerae* (6 nrMAGs), *Faecalibacterium sp900539945* (4 nrMAGs), *Dorea longicatena_B* (3 nrMAGs), *Blautia_A sp003480185* (3 nrMAGs), *Blautia_A sp003471165* (3 nrMAGs), *Dorea formicigenerans* (2 nrMAGs), *Fusicatenibacter saccharivorans* (2 nrMAGs), and *GCA-900066135 sp900543575* (2 nrMAGs). Importantly, we found that some of these species were previously reported (including in the original study Yeoh et al.) to be decreased in patients with COVID-19 such as *Blautia obeum*[11,19], *Faecalibacterium prausnitzii*[18,19,41,49], and *Dorea formicigenerans*[18,19]. Notably, multiple protective species (e.g., *Bariatricus comes*, *Blautia_A obeum*, *Blautia_A wexlerae*, *Dorea formicigenerans*, *Faecalibacterium prausnitzii_D*, *Faecalibacterium sp900539945*, and *Fusicatenibacter saccharivorans*) lost many strains in COVID-19 patients when compared to Non-COVID-19 controls (Fig. 4c, d). And the strain richness of most protective microbial species (17/21) negatively correlated with disease severity (Supplementary Data 1). Interestingly, those protective nrMAGs also showed a similar abundance distribution between patients with COVID-19 and Non-COVID-19 controls in the study of Zuo et al.[18] (Fig. S18). This finding provides strain level evidence that gut microbial taxa may interact with SARS-COV-2 infection and play a potential role in disease onset and progression in COVID-19.

## Genome annotation reveals functional differentiation between permissive and protective nrMAGs of COVID-19

To understand how those permissive and protective nrMAGs identified from the study of Yeoh et al.[19] may interact with SARS-CoV-2 infection, we next investigated whether the functional capacity of permissive and protective nrMAGs differ. To achieve that goal, we first annotated the genomes of permissive and protective nrMAGs using Prokka[50]. Then we processed the translated coding sequences using MicrobeAnnotator[51] for the functional annotation and calculated the KEGG module completeness (see Methods). Here, KEGG modules are functional gene units, which are linked to higher metabolic capabilities, structural complexes, and phenotypic characteristics[51]. A total of 231 and 254 KEGG modules were covered by at least one genome from permissive and protective nrMAGs, respectively. Principal component analysis revealed quite different metabolic potentials between permissive and protective nrMAGs (Fig. 7a, PERMANOVA: *P*-value = 0.0001). The main KEGG modules (with at least 50% module completeness) of each nrMAG are summarized in Fig. S19. Notably, we identified a set of KEGG modules that differed significantly in their module completeness between permissive and protective nrMAGs (Fig. 7b). For example, permissive nrMAGs showed significantly higher completeness level at the pentose phosphate pathway (e.g., M0004 and M0006) compared to protective nrMAGs. Moreover, we found multiple microbial genomes have the potential to use this pathway (Figs. S20, 21).

To further validate the association between the pentose phosphate pathway and COVID-19, we performed functional profiling for the metagenomics sequencing samples from the two discovery cohorts with case-control experimental settings (i.e., Zuo et al.[18] and Yeoh et al.[19]) as well as the three validation cohorts (i.e., Zhang et al.[41], Xu et al.[39], and Li et al.[22]) at the community level using HUMAnN3[52]. Notably, we found that the abundance of pentose phosphate pathway (PENTOSE-P-PWY) in COVID-19 patients was significantly higher than that in the Non-COVID-19 controls of the two discovery cohorts: Zuo et al. (Fig. S22a) and Yeoh et al. (Fig. S22b). This result is consistent with the result that the permissive nrMAGs showed significantly higher completeness level at the pentose phosphate pathway compared to protective nrMAGs. Moreover, we found that the pentose phosphate

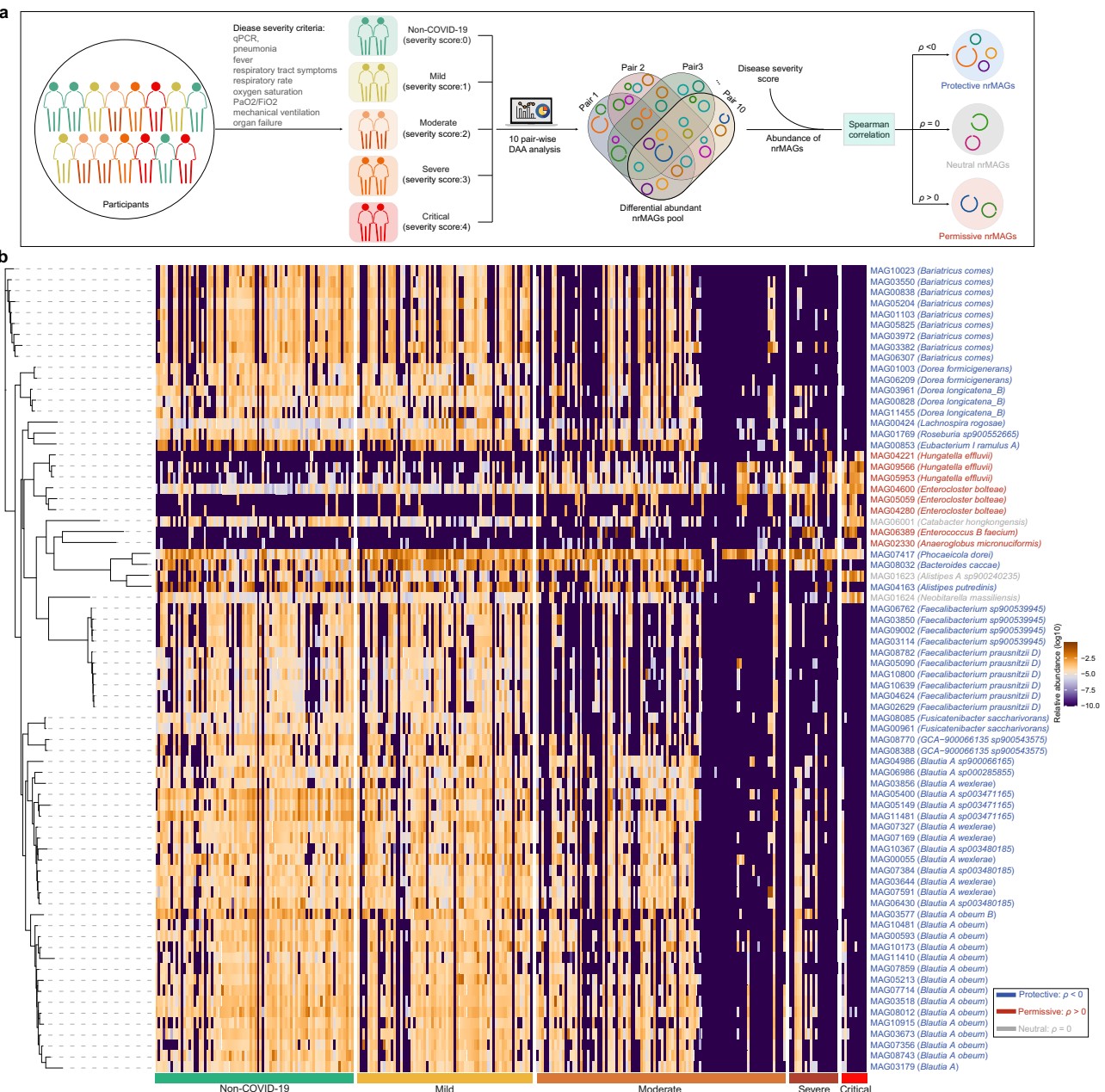

**Fig. 6 | The permissive and protective nrMAGs of COVID-19 identified by GMPT pipeline. a** The workflow of GMPT pipeline. The microbiome samples from the study of Yeoh et al. grouped to five groups (i.e., healthy control, mild, moderate, severe, and critical) based on disease severity. Differential abundance analysis was carried out on each possible pairwise comparison (10 pair-wise comparisons among five groups). The differentially abundant nrMAGs pool were originated from these pairwise analyses. The differentially abundant nrMAGs were ranked based on their frequency appearing in all the pairwise comparisons and differentiability (descending order). We further used the average relative abundance of nrMAGs and the disease severity to calculate the Spearman correlation coefficient. Here positive (negative) Spearman correlation coefficient ($\rho$) represent the permissive (protective) nrMAGs of COVID-19 severity. And the spearman correlation with 0 means the nrMAGs may be neutral to the severity of SARS-CoV-2 infection. **b** The heat map showed the abundance distribution of permissive, neutral, and protective nrMAGs across different disease severity groups identified using GMPT. These nrMAGs were taxonomically annotated using GTDB-Tk based on the Genome Taxonomy Database. The colors of the taxonomical label represent permissive, protective, or neutral nrMAGs. The phylogenetic tree of these nrMAGs was constructed using PhyloPhlAn.

pathway also showed higher abundance in COVID-19 patients from Zhang et al. ($p$-value = 0.39, Fig. S22c) and Xu et al. ($p$-value = 0.013, Fig. S22d) in validation cohorts.

## Discussion
Here, we leveraged hundreds publicly available WMS sequenced samples from multiple SARS-COV-2 datasets and generated for the first time a high-quality COVID-19 related genome catalog of the human microbiome. We recovered a large genome catalog representing 11,584

MAGs and 5403 nrMAGs of the human microbiome. Through the construction of this microbial genome catalog, we were able to provide the strain-level perspective to understanding the human microbiome and COVID-19.

By interrogating the WMS sequencing data with different technical settings, we gained a more comprehensive view of the microbial community associated with COVID-19. Due to the inherent differences (e.g., age, diet, and genetic background) across the different datasets, our goal was not targeted comparisons across datasets. Importantly,

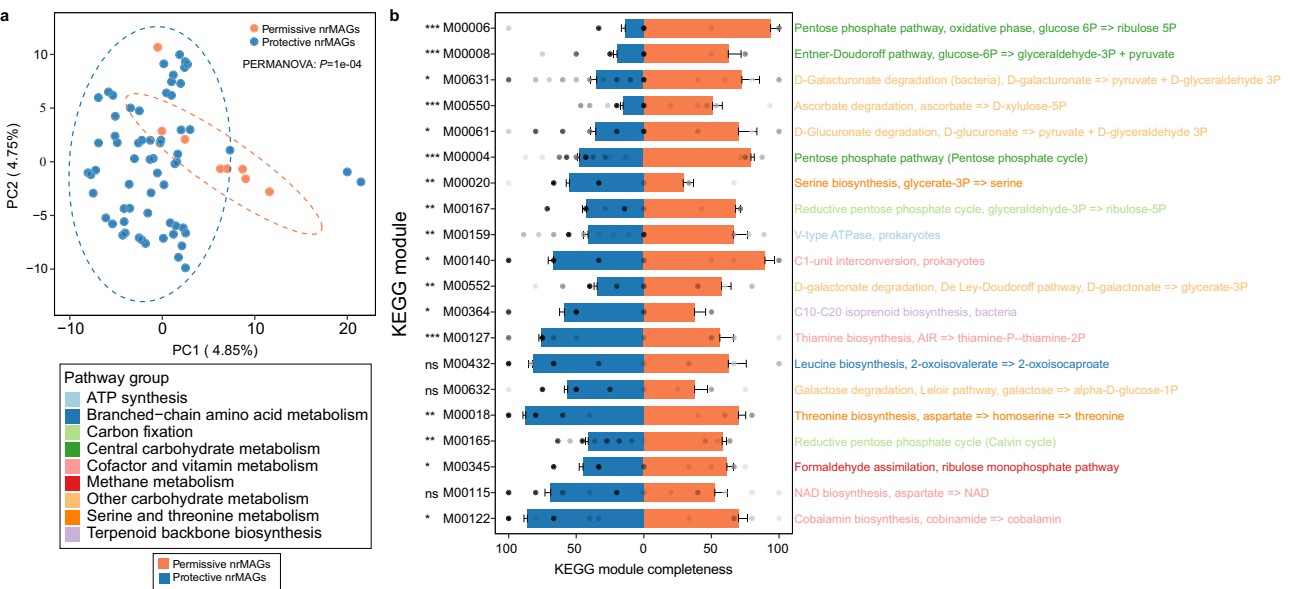

**Fig. 7 | Genome annotation of permissive and protective nrMAGs of COVID-19.** **a** Principal Component Analysis (PCA) plot of KEGG module completeness from all genomes of permissive ($n = 8$) and protective ($n = 63$) nrMAGs. PERMANOVA test was performed with 9999 permutations, two-sided. **b** The top 20 differential KEGG modules between permissive ($n = 8$) and protective ($n = 63$) nrMAGs ranked based on the mean difference. Data are presented as mean ± standard error of mean. $P$-values were calculated by two-sided Wilcoxon–Mann–Whitney test (ns non-significant, *$P < 0.05$; **$P < 0.01$; ***$P < 0.001$).

our study mainly focused on two discovery cohorts (Zuo et al.[18] and Yeoh et al.[19]) and three validation cohorts (Zhang et al.[41], Xu et al.[39], and Li et al.[22]) with well-defined case and control subjects. Although two nasopharyngeal microbiome datasets contained both patients with COVID-19 and Non-COVID-19 controls, the statistical power was limited by the small sample size (Liu et al.[36]) and a large portion of sequencing reads from another dataset (PRJNA743981[37]) were from the human host. Our analysis did not exclude these nasopharyngeal microbiome samples as they did contribute unique high-quality MAGs to our nrMAGs collection. Furthermore, two gut microbiome datasets (Britton et al.[53] and Cao et al.[54]) without Non-COVID-19 controls served as key resources of our MAGs collections.

Previous efforts have linked human microbiome diversity and COVID-19[11,14,34]. Coherently, the human gut microbiome of patients with COVID-19 in our study exhibited overall decreased alpha diversity at the nrMAGs level compared to the Non-COVID-19 healthy controls. Specifically, the Richness (number of nrMAGs) of the gut microbiome showed a relatively more consistent difference between COVID-19 and Non-COVID-19 samples in both discovery and validation cohorts. Notably, we found for the first time that COVID-19 patients lost many strains (nrMAGs) for certain microbial species when compared to Non-COVID-19 controls in both the discovery and validation cohorts. These findings suggest that SARS-CoV-2 infection is associated with a decrease in total strains that may be contributed by specific species. Moreover, some species lost multiple strains were also identified as protective microbial species by the GMPT pipeline. Interestingly, our analysis identified that patients with COVID-19 after recovery (negative for SARS-CoV-2 via RT-qPCR) differed more from Non-COVID-19 controls compared to patients with COVID-19 before recovery. This finding supports the possibility that the gut microbiome of patients with COVID-19 may not return to a relatively healthy status right after their recovery from COVID-19[55]. Given the fact that many patients recovering from SARS-CoV-2 infection have experienced prolonged COVID-19 symptoms[56], we hypothesize that long-lasting disease symptoms may be associated with changes in the gut microbiome but this needs to be explored further.

Using machine learning models, we demonstrated that the gut microbiome signatures at the nrMAG-level can accurately detected

COVID-19 from healthy controls. The high diagnostic accuracy of our microbiome-derived signature suggests that key microbial strains within the signature might play important roles in the pathogenesis of COVID-19. For example, some of nrMAGs at higher taxonomic levels (e.g., genus and species) have been reported to be correlated with COVID-19, such as *Blautia*[11,15,19], *Faecalibacterium prausnitzii*[18,19,49], and *Adlercreutzia equolifaciens*[19,22]. We also demonstrated that the gut microbiome signature identified from a specific discovery cohort can diagnose COVID-19 across separated cohorts, independent of the effects of host genetics and environmental factors on the gut microbiome. The universal nature of our microbiome-derived signature suggests that some key microbial species might play very important roles in the pathophysiology of SARS-CoV-2 infection. Notably, this study sheds important light on the ability of nrMAGs to predict the date of negative RT-qPCR result of patients with COVID-19. This analysis linked several microbial species from our important nrMAGs to the progression of COVID-19 such as *Citrobacter freundii*, *Veillonella parvula*, and *Parabacteroides distasonis*. Indeed, these species have been previous reported involved in COVID-19. For example, *Citrobacter freundii* was found to be significantly enriched in COVID-19 patients with fever[57]. *Veillonella parvula*[19,42,54] and *Parabacteroides distasonis*[19,58] were also shown to be a shared signature of COVID-19 in multiple studies. Importantly, we observed some opportunistic pathogens were associated with the progression of COVID-19, including nrMAGs from *Klebsiella quasivariicola*[43], *Klebsiella pneumoniae*[44], and *Escherichia coli*[45]. Related to our findings, multiple studies revealed high prevalence of bacterial pathogens in patients with COVID-19[18,59-62], further supporting the possibility that secondary infections by opportunistic pathogens may affect the progression of COVID-19. However, further studies are needed to validate these findings and determine how those microbes influence the progression of COVID-19.

In particular, host gut microbiota provides colonization resistance against pathogens, for example, a previous study reported that mice treated with neomycin antibiotics were more susceptible than control mice to influenza viruses[63]. And it turned out that neomycin-sensitive bacteria naturally present in the mice's bodies provided a trigger that led to the production of T cells and antibodies that could fight an influenza infection in the lungs. Notably, our study identified a set of

COVID-19 related nrMAGs and their determinants (i.e., permissive and protective) potentially involved in disease pathogenesis. It is important to note that these protective bacteria have also been reported to be related to SARS-CoV-2 infection. For example, *B. obeum* (a bacterial symbiont beneficial to the host immunity) was identified to be depleted in patients with COVID-19 in multiple studies[11,18]. *F. prausnitzii*, an anti-inflammatory and butyric acid-producing commensal bacterium, was found to be underrepresented in patients with COVID-19[18,19,64]. Moreover, our analysis enables the direct microbial genome comparison between permissive and protective nrMAGs. Consistent with previous reports of a relationship of the pentose phosphate pathway and SARS-CoV-2 infection[65–67], our findings support that overrepresentation of permissive nrMAGs and underrepresentation of protective nrMAGs may upregulate the pentose phosphate pathway as their genome are shown to be highly intact in those relevant modules. In addition to the genome annotation of permissive and protective nrMAGs, we also found that the overall abundance of pentose phosphate pathway in COVID-19 patients was higher than that in the Non-COVID-19 controls in two discovery cohorts and two validation cohorts. The pentose phosphate pathway is an important physiological process that can occur in 2 phases: oxidative and nonoxidative. Reactions of the pentose phosphate pathway, occur virtually ubiquitously, and maintain a central metabolic role in providing the RNA backbone precursors ribose 5-phosphate and erythrose 4-phosphate as precursors for aromatic amino acids[68]. The aromatic amino acids in the juxtamembrane domain of the SARS-CoV S glycoprotein play critical roles in receptor-dependent virus-cell and cell-cell fusion[69]. A previous study reported that the UK mutation (N501Y: a mutation from asparagine to tyrosine conferring one more aromatic amino acid to receptor binding domain) interacts closely with Y41 (ACE2) in the receptor therefore producing aromatic-aromatic interactions that provide for stronger binding between receptor and spike[70]. Indeed, the levels of aromatic amino acids (e.g., tyrosine, phenylalanine, and tryptophan) were increased significantly in COVID-19 patients compared with controls using targeted metabolic analysis[71].

Importantly, a previous study reported a significant increase in the levels of some intermediates of the glycolytic and pentose phosphate pathways in sera of COVID-19 positive patients[65]. Moreover, an earlier study (86 COVID-19 patients and 57 healthy controls, United Arab Emirates) reported that the pentose phosphate pathway was significantly upregulated on COVID-19 patient microbiome samples using 16 S rRNA gene sequencing together with a phylogenetic investigation of communities by reconstructing unobserved state (PICRUSt)[72]. In addition, SARS-CoV-2 infection was found to be associated with changes in the regulation of the pentose phosphate pathway in both in vivo (Caco-2 cells)[66] and in vitro (ferret model)[67] studies. Together, these results suggest that specific microbes (permissive nrMAGs, such as strains from *Hungatella effluvii* and *Enterocloster bolteae*) may play a role in mediating SRAS-CoV-2 entry into host cells through pentose phosphate pathway and aromatic amino acids. However, further mechanistic studies are warranted to test the exact role of our candidate permissive and protective nrMAGs in SARS-CoV-2 infection.

The current study has several limitations. First, although we included a large number of shotgun metagenomic sequencing samples from the COVID-19 related human microbiome study (publicly available as of August 2021 and April 2022 on the discovery and validation cohorts, respectively), most of the microbiome samples came from China. This limitation could be addressed by following this work with collection of more human microbiome samples from different populations and body sites to construct a more comprehensive genome catalog to reveal the full landscape of the human microbiome in COVID-19. Second, even though we adjusted for potential confounder in our statistical models, we were unable to assess some covariates

such as: medication, diet, and psychological stress that are not publicly available. Third, consistent with multiple MAG-related WMS studies[27,73,74], we only recovered MAGs from bacteria and archaea. Given the fact that de novo discovery of non-bacterial genomes is quite challenging[75], future study targets for other domains, including fungi and viruses, will give a more comprehensive view in the context of host-specific microbiotas and COVID-19. Although the majority of MAGs we reconstructed in this study have high quality, future investigations aiming for recovering the complete genome of microbes will further enhance our understanding of the interaction between human microbiome and SARS-CoV-2 infection[76,77]. Finally, additional experiments are needed to assess the casual role of candidate permissive and protective nrMAGs in COVID-19 progression. Nevertheless, given the large uncultured diversity still remaining in the human gut microbiome and deficiency of both annotated genes and reference genomes, having a high-quality genome catalog substantially enhances the resolution and accuracy of metagenome-based COVID-19 studies. Therefore, the presented genome catalog represents a key step toward mechanistic understanding the role of the human gut microbiome in SARS-CoV-2 infection.

In summary, we present here the first construction of the genome catalog using assembly and reference free binning of metagenome in patients with COVID-19 and Non-COVID-19 controls. Our findings support the close connection between SARS-CoV-2 infection and the human gut microbiome, and we demonstrate that the main findings of this study can be largely validated in independent cohorts. These insights into metagenomic strain-level aspects of relation in human microbiome and COVID-19 and genome context will form the basis of future studies.

## Methods
### Data collection
We identified COVID-19 metagenomic sequencing studies from keyword searches in PubMed and online repositories (i.e., NCBI, ENA, and GSA) and by following references in meta-analyses and related microbiome studies. We included samples with publicly available raw shotgun metagenomic sequencing data (paired fastq files) and metadata indicating patients with COVID-19 or Non-COVID-19 control status. All the sequencing data were downloaded from online repositories or links provided in the original publications, but some metadata were acquired after personal communication with the authors. We did not include any studies which required additional ethics committee approvals or authorizations for access. A total of 514 and 341 microbiome samples from six discovery cohorts and three validation cohorts were analyzed in this study, respectively (Table 1 and Table S1).

### Metagenome assembly and binning
Genome reconstruction of human microbiome with metagenomic sequencing data was performed with the function modules of metaWRAP (v1.3.2)[78], which is a pipeline that includes numerous modules for constructing metagenomic bins. First, the metaWRAP-Read_qc module was applied to trims the raw sequence reads and removes human contamination for each of the sequenced samples. Then the clean reads from the sequencing samples were assembled with the metaWRAP-Assembly module using metaSPAdes (v3.13.0)[79]. Thereafter, MaxBin2 (v2.2.6)[80], metaBAT2 (v2.12.1)[81], and CONCOCT (v1.0.0)[82] were used to bin the assemblies. The default of the minimum length of contigs used for constructing bins with MaxBin2 and CONCOCT were 1000 bp, and metaBAT2 was defaulted to 1500 bp[78]. Refinement of MAGs was performed by the bin_refinement module of metaWRAP[78], and CheckM (v1.0.12)[83] was used to estimate the completeness and contamination of the bins, and the minimum completion and maximum contamination were 50% and 10%, respectively.

## Species-level clustering and dereplication and of MAGs

All 11,584 MAGs were clustered into species-level genome bins (SGBs) at the threshold of 95% ANI using the 'cluster' program in dRep (v3.0.0)[84]. All MAGs were taxonomically annotated using GTDB-Tk (v.1.4.1)[85] based on the Genome Taxonomy Database (http://gtdb.ecogenomic.org/)[31], which produced standardized taxonomic labels that were used for the analysis in this study. SGBs containing at least one reference genome (or MAG) in the Genome Taxonomy Database were considered as known SGBs. And SGBs without reference genomes were considered as unknown SGBs (uSGBs)[75]. dRep (v3.0.0)[84] was then used for dereplication of all 11,584 MAGs (≥50% genome completeness and ≤5% contamination) by two-steps. First, MAGs were divided into primary clusters using Mash[86] at a 90% Mash ANI. Then, each primary cluster was used to form secondary clusters at the threshold of 99% ANI with at least 30% overlap between genomes. According to the criteria of quality evaluation by CheckM (v1.0.12)[83], 5403 nrMAGs were divided into medium-quality MAGs (50% ≤ completeness < 90% and ≤5% contamination) and high-quality MAGs (≥90% completeness and ≤5% contamination).

## Abundances estimation and phylogenetic analysis of nrMAGs

The metaWRAP-Quant_bins module integrated with Salmon[87] (v0.13.1) was used to estimate the abundance of each nrMAGs in each of the metagenomic samples (both the discovery and validation cohorts). The phylogenetic tree of the nrMAGs was built using PhyloPhlAn (v3.0.58)[88]. The tree was visualized using iTOL (https://itol.embl.de/)[89].

## Genome annotation of nrMAGs

The genome annotation of MAGs was first performed with Prokka (v1.13)[50] using the annotate_bins module of metaWRAP[78]. The annotated genomes were then processed with MicrobeAnnotator (v2.0.5)[51] for the functional annotation and to calculate KEGG module completeness. All proteins are searched against the curated KEGG Ortholog (KO) database using Kofamscan[90]; best matches are selected according to Kofamscan's adaptive score threshold. Proteins without KO identifiers (or matches) are extracted and searched against other databases (e.g., Swissprot, curated RefSeq database or non-curated trEMBL database)[51]. The KO identifiers associated with all proteins in each genome (or set of proteins) are extracted, and KEGG module completeness is calculated based on the total steps in a module, the proteins (KOs) required for each step, and the KOs present in each genome. Finally, the results were compiled in a single matrix-like module completeness table for all genomes.

## Functional profiling of metagenomic sequencing data

Functional profiling was performed using HUMANN3(v3.0.1)[52]. Briefly, for each microbiome sample, taxonomic profiling is used to identify detectable organisms. Reads are recruited to sample-specific pangenomes including all gene families in any detected microorganisms using Bowtie2 (v2.4.5). Unmapped reads are aligned against UniRef90 (v201901b) using DIAMOND (v2.0.15) translated search.

## Statistical analysis

Microbial alpha and beta diversity measures were calculated at the nrMAGs level using vegan package (v2.5.7) in R. Principal coordinates analysis (PCoA) plots were generated with Bray−Curtis dissimilarity. Differences in microbiome compositions across different groups were tested by the permutational multivariate analysis of variance (PERMANOVA) using the "adonis" function in R's vegan package. All PERMANOVA tests were performed with 9999 permutations based on Bray−Curtis dissimilarity. Differences between groups were analyzed using a Wilcoxon−Mann−Whitney test. For differential abundance analysis in GMPT (Generalized Microbe Phenotype Triangulation) pipeline[46], we used ANCOM (analysis of composition of microbiomes)[47], with a Benjamini−Hochberg correction at 5% level of significance, and adjusted each patient's identifier as a random effect. Only the nrMAGs that were presented in at least 5% of samples were included. The phylogenetic tree of the permissive, neutral, and protective nrMAGs was built using PhyloPhlAn (v3.0.58)[88] and then visualized using iTOL (https://itol.embl.de/)[89].

To develop a model capable of distinguishing patients with COVID-19 versus Non-COVID-19 controls, we implemented Random Forest using R' random Forest package. A custom machine learning process was conducted using features of nrMAGs with 5-fold cross validation. The data was split into a training set and a test set, with 80% of the data forming the training data and the remaining 20% forming the test set. And then we randomly split the data 50 times. The performance of the classification model was evaluated using AUROC (area under the receiver operating characteristic curve) and AUPRC (area under the precision-recall curve) on the test set. The importance of each feature was quantified by the Mean Decrease in Accuracy (MDA) of the classifier due to the exclusion (or permutation) of this feature. The more the accuracy of the classifier decreases due to the exclusion (or permutation) of a single feature, the more important that feature is deemed for classification of the data. We then built Random Forest regression model with 5-fold cross-validation to predict the exact date of microbiome sample collected before or after negative RT-qPCR result. We randomly split the data 50 times. The importance of each feature in the regression was quantified by percent increase in mean square error. Pearson correlation coefficient between the true and predicted date of negative RT-qPCR result was used to evaluate the performance. All statistical analysis was performed with R (version 3.6.3).

## Reporting summary

Further information on research design is available in the Nature Research Reporting Summary linked to this article.

## Data availability

All data used in this article come from publicly available sources. The metagenomic data from the discovery cohorts are available in the NCBI or Genome Sequence Archive Bioproject database under accession code PRJNA624223, PRJNA656660, PRJNA660883, PRJNA743981, PRJCA003532 (https://ngdc.cncb.ac.cn/gsa/browse/CRA003271), and PRJNA650244. The metagenomic data from the validation cohorts are available in the NCBI Bioproject or SRA database under accession code PRJNA689961, SRP118759, PRJNA792726, PRJEB43555. Metagenome-assembled genomes for all samples are available on Figshare (https://figshare.com/s/a426a12b463758ed6a54). HUMANN3 databases for metagenomic functional profiling were accessed from http://huttenhower.sph.harvard.edu/humann_data/.

## Code availability

The codes for construction of the MAGs catalog and statistical analyses and visualization are available in the GitHub repository (https://github.com/Owenke247/COVID-19) or the Zenodo database (https://doi.org/10.5281/zenodo.6824864).

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

## Acknowledgements

We would like to thank Dr. Yun Kit Yeoh and Dr. Siew C Ng for sharing the phenotypic data with us. We thank Xu-Wen Wang, Zheng Sun, Tong Wang, Darius Schaub, Yunyan Zhou, and Xiaochang Huang for valuable discussions. Yang-Yu Liu acknowledges the funding support from the National Institutes of Health (R01AI141529, R01HD093761, RF1AG067744, UH3OD023268, U19AI095219, and U01HL089856).

## Author contributions

S.K. and Y.-Y.L. conceived and designed the project. S.K. performed all the data analysis. S.K. and Y.-Y.L. interpreted the results and prepared the manuscript. S.T.W. reviewed and edited the manuscript. All authors approved the manuscript.

## Competing interests

The authors declare no competing interests.
