## [Peer Review File · Nature Communications]

REVIEWER COMMENTS

Reviewer #1 (Remarks to the Author):

This study investigates the potential role of human microbiome in COVID-19 infection, progression, and symptoms by combining 6 publicly available data sets (4 from Chinese population; 1 US; 1 unspecified), yielding 96 nasal and 418 fecal samples from 359 individuals, including 110 (~20%) control samples; the main focus is on two of these publicly available studies that have well-defined cases and controls. Altogether 5403 non-redundant strain-resolution MAGs are identified by mapping against GTDB after reference-free binning and metagenome assembly. Comparisons suggest that the resulting microbiome profiles can be used to classify cases and controls with a relatively good accuracy (AUPRC 0.88-0.97 in individual data sets) and some overlapping features. Opportunistic pathogens that have been associated with generally compromised health status in earlier studies are linked to disease progression (qPCR test results). The comparisons cover standard aspects of microbiome alpha & beta diversity, differential abundance, temporal stability, and functional analysis which suggests further role in the disease manifestation. The study contributes new knowledge on COVID-19 associated changes in nasal and fecal microbiomes. Data and code are available, statistical analyses has been performed appropriately and the conclusions are supported by the data.

The manuscript appears technically sound and well written and uses established state-of-the-art methodology; relevant literature has been cited and observations are supported by the earlier studies. I have mainly reviewed the statistical and bioinformatics aspects of the work.

I only have the following major remarks related to code and data availability.

1. I did not find a code availability statement. The software packages are mentioned; the source code of the data analysis workflow would be essential to assess the full technical details in the analyses. An open license and permanent DOI would be recommended.

2. The source data sets are publicly available. Is the newly assembled MAG catalogue also released openly together with the manuscript? I tried to search this information from the manuscript.

* Minor

line 538: PcoA -> PCoA

Reviewer #2 (Remarks to the Author):

In this study, ke et al delineated the construction of the genome catalog using assembly and reference-free binning of metagenome in patients with COVID-19 and Non-COVID-19 controls. The manuscript is well-written and the analysis is sophisticated. However, several major limitations hamper the impact of this study.

Major concerns:

1. Overall, this study did not report too many distinct findings from prior reports although they adopted a new strategy to analyze WMS sequencing data. Moreover, this strategy is not a method developed by themselves and has been used in other studies in recent years. Therefore, the novelty of the work is not very high.
2. It is still a descriptive study without mechanistic insights and is unable to dissect the causality between gut microbiota and COVID-19, thereby limiting the significance of the study.
3. They analyze metagenomic data from 514 samples, while only two data sets (Zuo et al and Yeoh et al) are mainly used. Since emerging studies on fecal metagenomes of patients with COVID-19 are public available now, it is nice to include more data sets to validate the findings.
4. Line 413-414: It is unclear which statistical models had adjusted the confounders and what kinds of confounders have been adjusted. This needs to be clarified in methods or results. Moreover, the lack of comprehensive clinical information and assessment of covariates may affect the robustness and generalization of this study.
5. Lines 298-320: The authors showed a significantly higher completeness level at the pentose phosphate pathway compared to protective nrMAGs. Did they check whether the relative

abundance of pentose phosphate pathway is higher in patients with COVID-19 compared with controls in their cohort, and which strain has the potential to use this pathway?

Minor concerns:

1. Figure 3: The sample size of each group (each bar) needs to be clarified in the figure or figure legend.

2. Figure 5: The color bar indicating the Spearman correlation coefficient is unable to be found in Figure 5b.

Reviewer #3 (Remarks to the Author):

This work describes the analysis of whole-metagenome shotgun sequencing data with reconstruction of metagenome-assembled genomes (MAGs) from COVID and non-COVID patients from several Chinese cohorts. The study includes both fecal and nasopharyngeal samples.

The authors focus most of their analyses on 1-2 cohorts that include control individuals and therefore the power of the study is reduced compared to what initially the reader would think.

Several concerns arise from reading this manuscript.

First of all, the attempt to have microbiome signatures from COVID and non-COVID types is not convincing. In fact, several species associate with one or the other phenotype but such species vary depending on the cohort, the results between datasets are not consistent.

The diversity difference is only explored at indexes level. Since the authors also have MAGs that are resolved at strain level, it could be interesting to understand whether there's strain-level diversity within the same species according to covid vs non-covid phenotypes (or even with severity stratification). In other words, how many strains of the same species are found in covid and non-covid cases? Is there a strain dominance in species associated to the covid subjects?

Finally, the functional metagenomics part is potentially interesting but not enough exploited or interpreted.

The discussion lacks of proper results' interpretation and is in some paragraphs too repetitive of the results section.

The study appears overall preliminary and the proposed microbiome signature of COVID disease and progression are not convincing.

Specific comments:

168-174. Please consider erasing this section as it is redundant compared to fig. 2d. Moreover, these are the common phyla human microbes belong to.

176-192. Looking at microbial diversity in COVID and non COVID patients is surely interesting. However, the results show no clear trend of differences. The choice of separating the studies is arguable, the reader would think that considering all subjects together there are no differences?

184. This is not true for the Zuo cohort.

194-199. I still don't see a clear difference here. It appears that non-covid patients are more similar while there's an increased variability in covid individuals, this is the result I see here.

In addition, Fig.3d,e,g,h could be left out.

223-245. The fact that the COVID signature species are different between datasets makes the COVID signature concept and the possibility to have COVID microbiome markers quite weak.

298-320. Are these also taken from 1 dataset only? In addition, also in the case of functional potential, the associations with permissive and protective cases seems quite random and not well explained.

399-405. These results do not indicate a clear set of metabolic pathways that associate with a permissive or protective signature and that can explain the development of the covid disease. There is no clear idea on how the gut microbes can interact with the virus infection or how they can change their metabolic potential with progression of the virus.

Responses to Reviewer #1

This study investigates the potential role of human microbiome in COVID-19 infection, progression, and symptoms by combining 6 publicly available data sets (4 from Chinese population; 1 US; 1 unspecified), yielding 96 nasal and 418 fecal samples from 359 individuals, including 110 (~20%) control samples; the main focus is on two of these publicly available studies that have well-defined cases and controls. Altogether 5403 non-redundant strain-resolution MAGs are identified by mapping against GTDB after reference-free binning and metagenome assembly. Comparisons suggest that the resulting microbiome profiles can be used to classify cases and controls with a relatively good accuracy (AUPRC 0.88-0.97 in individual data sets) and some overlapping features. Opportunistic pathogens that have been associated with generally compromised health status in earlier studies are linked to disease progression (qPCR test results). The comparisons cover standard aspects of microbiome alpha & beta diversity, differential abundance, temporal stability, and functional analysis which suggests further role in the disease manifestation. The study contributes new knowledge on COVID-19 associated changes in nasal and fecal microbiomes. Data and code are available, statistical analyses has been performed appropriately and the conclusions are supported by the data.

The manuscript appears technically sound and well written and uses established state-of-the-art methodology; relevant literature has been cited and observations are supported by the earlier studies. I have mainly reviewed the statistical and bioinformatics aspects of the work.

We thank Reviewer #1 very much for the thorough evaluation of our work and very positive assessment. We next address each of the reviewer's comments in order.

I only have the following major remarks related to code and data availability.

1. I did not find a code availability statement. The software packages are mentioned; the source code of the data analysis workflow would be essential to assess the full technical details in the analyses. An open license and permanent DOI would be recommended.

We thank Reviewer #1 for this critical comment. We apologize for not explicitly providing a code availability statement in the previous submission. (It was included in the Report Summary Checklist, but not in the manuscript itself.) We have now added it to the revised manuscript (see **main text: page 17, lines 660-663**).

“The codes for construction of the MAGs catalog and statistical analyses and visualization are available in the GitHub repository (<https://github.com/Owenke247/COVID-19>) or the Zenodo database (<https://doi.org/10.5281/zenodo.6824864>).”

2. The source data sets are publicly available. Is the newly assembled MAG catalogue also released openly together with the manuscript? I tried to search this information from the manuscript.

We thank Reviewer #1 for pointing this out. We have now uploaded the MAGs to an online open access repository Figshare and added the following statement in the revised manuscript (see **main text: page 17, lines 665-671**).

“Metagenome-assembled genomes for all samples are available on Figshare (<https://figshare.com/s/a426a12b463758ed6a54>).”

* Minor

line 538: PcoA -> PCoA

We thank Reviewer #1 for catching this typo. We have replaced “PcoA” by “PCoA” in the revised manuscript (see **main text: page 16, line 630**).

Finally, we would like to thank Reviewer #1 again for reviewing our manuscript. We hope our responses above have addressed all the comments in a satisfactory manner.

Responses to Reviewer #2

In this study, Ke et al delineated the construction of the genome catalog using assembly and reference-free binning of metagenome in patients with COVID-19 and Non-COVID-19 controls. The manuscript is well-written and the analysis is sophisticated. However, several major limitations hamper the impact of this study.

We thank Reviewer #2 for reviewing our manuscript. We next address each of the reviewer's concerns in order.

Major concerns:

1. Overall, this study did not report too many distinct findings from prior reports although they adopted a new strategy to analyze WMS sequencing data. Moreover, this strategy is not a method developed by themselves and has been used in other studies in recent years. Therefore, the novelty of the work is not very high.

We thank Reviewer #2 for this critical comment.

We fully agree with the reviewer that our analysis is based on publicly available datasets and existing methods. Hence, we didn't claim the novelty of our work in terms of **methodology development** or **microbiome data collection**. But we do want to emphasize the novelty of our work in terms of its **key findings**, as listed here.

- 1) This work led to, for the first time, the construction of COVID-19 related microbial genome catalog and identified 160 novel species-level genome bins (SGBs), which have no available genome related information in the Genome Taxonomy Database (GTDB).
- 2) This work led to the strain-level resolution of the COVID-19 related microbial communities. Most previous studies on the human microbiome of COVID-19 patients only provided the taxonomic resolution at the species (or even genus) level. Up to our knowledge, so far there have been only two COVID-19 related microbiome studies that offered strain-level resolution. One study¹ is based on the single nucleotide variant (SNV) similarity and the patterns of presence/absence of selected KEGG metabolic pathways. The other study² is based on PathoScope2.0³, a reference-based computational framework for strain identification in microbiome samples. Neither approach is based on a combination of *de novo* metagenomic assembly and binning.
- 3) Remarkably, in responding another reviewer's comment, we found that COVID-19 patients lost many strains for certain microbial species when compared to Non-COVID-19 controls (**Fig.R1-R4**). This finding is completely novel and has been added to the revised manuscript (see **main text: Fig.4** and **SI: FigS.10-12**).
- 4) The genomic content of MAGs at the strain level presented here provides a unique resource to inform microbiome-based therapeutic developments for COVID-19 progression and post-COVID conditions. For example, the genomic content of a specific strain will help researchers isolate the target microbes through direct genome comparison.

2. It is still a descriptive study without mechanistic insights and is unable to dissect the causality between gut microbiota and COVID-19, thereby limiting the significance of the study.

We thank Reviewer #2 for this critical comment.

First of all, we emphasize that in this work we have tried to move beyond correlations and began to address causation. As we know, microbiome-wide association studies (MWAS) have been widely applied to identify phenotype associated microbes, however these studies typically generate a long list of commensals implicated as biomarkers of disease, with no clear relevance to disease pathogenesis. Recently, the so-called microbe-phenotype triangulation (MPT) method⁴, and its generalization --- the generalized microbe-phenotype triangulation (GMPT)⁵ have been developed to identify potentially causal microbes that influence disease pathogenesis. Both methods compare the microbial communities of groups of hosts that elicit divergent phenotypes (e.g., susceptibility to disease) to pinpoint causal microbes. The key assumption of the MPT method is that if any taxa were causal to disease pathogenesis, they would be present in all pair-wise comparisons of different phenotype groups. Potentially causal taxa obtained from MPT typically are at a high taxonomic level, e.g., the family level. Species/strains from these taxa then have to be tested individually to determine whether they play any causal roles in determining the observed host phenotype. GMPT overcomes the limitation of MPT by assuming that those differentially abundant species/strains that appear in most of the pair comparisons and whose abundances display a strong negative or positive correlation with the disease severity are strong candidates for preventive or permissive species/strains. GMPT enables us to move beyond correlation analyses and address causation in an efficient way.

Second, those protective nrMAGs identified by applying GMPT to the data of Yeoh et al. showed a similar abundance distribution between patients with COVID-19 and Non-COVID-19 controls in the study of Zuo et al. (see **SI: Fig.S18**). More importantly, for multiple identified protective nrMAGs (strains), their corresponding species (e.g., *Blautia obeum* and *Faecalibacterium prausnitzii*) have been reported to be related to SARS-CoV-2 infection. For example, previous studies have shown that: (1) *B. obeum* (a bacterial symbiont beneficial to the host immunity) is significantly reduced in COVID-19 patient⁶⁻⁸; (2) *F. prausnitzii* (an anti-inflammatory and butyric acid-producing commensal bacterium) is underrepresented in COVID-19 patients⁷⁻¹². Therefore, our prediction on the protective nature of those nrMAGs is consistent with previous studies.

Finally, we fully agree with the reviewer that further mechanistic studies are warranted to assess the exact casual role of our predicted permissive and protective nrMAGs in COVID-19. Those mechanistic studies, regardless of their preclinical or clinical nature, are unfortunately beyond the scope of the current work of computational nature. But we believe that the results presented in our computational study will inform subsequent mechanistic studies.

3. They analyze metagenomic data from 514 samples, while only two data sets (Zuo et al and Yeoh et al) are mainly used. Since emerging studies on fecal metagenomes of patients with COVID-19 are public available now, it is nice to include more data sets to validate the findings.

We thank Reviewer #2 for this very constructive comment.

Following the reviewer’s suggestion, we have identified 9 additional COVID-19 metagenomic sequencing studies from PubMed and other online repositories (i.e., NCBI, ENA, and GSA, published as of April 2022). Here is a table brief summarizing the 9 datasets:

Study	COVID-19 positive samples	COVID-19 negative samples	COVID-19 positive subjects	COVID-19 negative subjects	Sample Type	Geography	Year	Availability of raw sequencing data	Notes
Hazan et al.¹³	50	20	50	20	Fecal	USA	Mar-22	No	Raw sequencing data not publicly available
Liu et al.⁸	177	68	106	68	Fecal	CHINA	Jan-22	Yes	PRJNA714459, study design is quite different.
Li et al.¹⁴	46	19	46	19	Fecal	CHINA	Oct-21	Yes	PRJEB43555
Sun et al.¹⁵	63	8	63	8	Fecal	CHINA	Jan-22	No	Raw sequencing data not publicly available
Venzon et al.¹⁶	130	0	96	0	Fecal	USA	Mar-22	No	Raw sequencing data not publicly available.
Xu et al.¹⁷	38	31	38	31	Fecal	CHINA	Mar-22	Yes	SRP118759, PRJNA792726
Zhang et al.¹⁸	129	78	66	70	Fecal	CHINA	Feb-22	Yes	PRJNA689961
Bai et al.¹⁹	37	20	37	20	Nasopharyngeal	Sweden	Apr-22	Yes	PRJNA781460
Sulaiman et al.²⁰	118	0	118	0	Nasopharyngeal	USA	Oct-21	No	PRJNA687506

From the 9 datasets, we first excluded the datasets without publicly available raw shotgun metagenomic sequencing data (i.e., paired fastq files). We then excluded the datasets that do not contain metadata indicating patients with COVID-19 or Non-COVID-19 control status (i.e., Hazan et al., Sun et al., and Venzon et al.). Given that our analysis focused primarily on the gut microbiome data, two studies of the nasopharyngeal samples (i.e., Bai et al. and Sulaiman et al.) were excluded from further analysis. In the study of Liu et al., they investigated the long-term associations between the gut microbiome composition and persistent symptoms in patients with COVID-19 up to 9 months after the clearance of SARS-CoV-2 virus. Due to its quite different study design from other studies, we excluded this dataset from further analysis. We finally chosen

to analyze the data from studies of Zhang et al., Xu et al., and Li et al. as external **validation cohorts** (colored in green in the above table).

To validate our key findings in the **discovery cohorts** (i.e., Zuo et al. and Yeoh et al.), we first calculated the nrMAG (strain) abundance profiles of microbiome samples in the validation cohorts following the same pipeline (i.e., directly mapped the clean sequencing data to our high-quality microbial genome catalog constructed from the discovery cohorts) as used for the analysis of the discovery cohorts. We investigated if SARS-CoV-2 infection is associated with alpha diversity on the validation cohorts at the nrMAG-level. The alpha diversity measures (i.e., Richness and Shannon index) from COVID-19 patients and Non-COVID-19 controls were compared. In accordance with the discovery cohorts, we found that the alpha diversity of the gut microbiome in COVID-19 patients was overall lower than that in Non-COVID-19 controls in the validation cohorts (**Fig.R5**). We did not observe significant differences in alpha diversity between COVID-19 patients and Non-COVID-19 controls on the dataset of Xu et al. This might be due to the fact that the microbiome samples of COVID-19 patients (collected on 2020) and Non-COVID-19 controls (collected on 2016) were not collected and sequenced at the same time¹⁷.

Notably, we found that COVID-19 patients lost many strains for certain microbial species when compared to Non-COVID-19 controls for both the discovery (**Fig.R1**) and validation cohorts (**Fig.R3**). In line with the discovery cohorts, COVID-19 patients in two datasets (Xu et al. and Li et al) showed significantly higher within-group variation than their Non-COVID-19 controls (**Fig.R6**).

To evaluate the universal utility of the gut microbiome signature for the detection of COVID-19, we tested its diagnostic accuracy on the three validation cohorts. Briefly, we trained our machine learning model using samples from the cohort of Yeoh et al. (which has the largest sample size) and then test it on the 3 validation cohorts. We found that our machine learning model achieved an overall classification performance with AUROC~0.979 and AUPRC~0.976 in the cohort of Zhang et al, respectively (**Fig.R7a**); AUROC~0.660 and AUPRC~0.571 in the cohort of Xu et al. (**Fig.R7b**); AUROC~0.699 and AUPRC~0.565 in the cohort of Li et al. (**Fig.R7c**). The performance differences on the validation cohorts may be due to the fact that the cohort of Zhang et al. is created by the same research group as that of Zuo et al. and Yeoh et al., while the cohorts of Xu et al. and Li et al. were created by different groups. We know that different research groups may have different protocols for handling microbiome samples. Indeed, we found multiple differences in the microbiome protocols between the discovery cohorts and validation cohorts in terms of sample collection (e.g., collected by hospital staff or self-collected and collection tubes containing preservative media), sample transplantation (e.g., ice pack, liquid nitrogen, and collection tubes containing preservative media), DNA extraction (e.g., sample volume: 0.1g or 0.2g, preprocessing: ddH₂O or no pretreatment, and DNA collection kit: QIAamp Fast DNA Stool Mini Kit or Maxwell RSC PureFood GMO and Authentication Kit), sequencing platform (e.g., Illumina NovaSeq 6000 and BGISEQ-500 platform), etc. Hence, it is not a big surprise that a machine learning model trained on microbiome data created by one research group does not work very well on a dataset created by another research group. This actually highlights the importance of protocol standardization in microbiome studies²¹.

Due to the lack of data on disease severity (i.e., healthy controls, mild, moderate, severe, and critical) in those validation cohorts, unfortunately we were unable to validate the results of our GMPT pipeline further in identifying potentially causal microbes that influence disease pathogenesis.

Overall, through three external validation cohorts, we found that the main findings from the discovery cohorts could be reproduced in those independent cohorts. We believe that the power of our study has been significantly improved. The above results have been added to the revised manuscript (see **main text: Fig.4**; and **SI: Figs.S7, S9-S12, S16, and S22**).

4. Line 413-414: It is unclear which statistical models had adjusted the confounders and what kinds of confounders have been adjusted. This needs to be clarified in methods or results. Moreover, the lack of comprehensive clinical information and assessment of covariates may affect the robustness and generalization of this study.

We thank Reviewer #2 for raising this very important point.

First, we fully agree with the reviewer that clinical information and other covariates are important in human microbiome studies. Given that potential covariates such as medication, diet, and psychological stress have not been released by the authors of those studies, unfortunately we were unable to consider those factors in our statistical models. We acknowledged that this is a limitation in our current study.

Second, we emphasize that the availability of those clinical information and covariates will not affect the final MAG catalog, which is a key result of our current study.

Third, our GMPT pipeline is quite robust to those clinical information and covariates. In the Methods section of the previous manuscript, we did mention that the subject ID was treated as a random effect in the GMPT pipeline. In the revised manuscript, this statement is now included in the Results section (see **main text: page 9, line 335**). Moreover, we found that those COVID-19 related nrMAGs identified by applying GMPT to the data of Yeoh et al. showed a similar abundance distribution between COVID-19 patients and Non-COVID-19 controls in the study of Zuo et al (see **SI: Fig.S18**), which suggest high generalizability of our results.

Finally, we can largely validate our key results using the three validation cohorts, which suggests the robustness and generalizability of our results.

5. Lines 298-320: The authors showed a significantly higher completeness level at the pentose phosphate pathway compared to protective nrMAGs. Did they check whether the relative abundance of pentose phosphate pathway is higher in patients with COVID-19 compared with

controls in their cohort, and which strain has the potential to use this pathway?

We thank Reviewer #2 for this critical comment.

Following this suggestion, we performed functional profiling for the microbiome samples of the two discovery cohorts with case-control experimental setting (i.e., Zuo et al. and Yeoh et al.) as well as three validation cohorts (i.e., Zhang et al, Xu et al., and Li et al.), using a well-accepted function profiler in human microbiome studies: HUMAnN3²².

Notably, we found that the abundance of pentose phosphate pathway (PENTOSE-P-PWY) in COVID-19 patients was significantly higher than that in Non-COVID-19 controls of the two discovery cohorts: Zuo et al. (**Fig.R8a**) and Yeoh et al. (**Fig.R8b**). Among the three validation cohorts, we found that the pentose phosphate pathway showed higher abundance in COVID-19 patients from Zhang et al. (p value=0.39, **Fig.R8c**) and Xu et al. (p value=0.013, **Fig.R8d**). However, for the validation cohort (Li et al., which is the smallest cohort among the three validation cohorts), we found the opposite result (**Fig.R8e**). This inconsistency may be due to the inherent differences between cohorts such as sample collection, DNA extraction, sequencing platform, etc. The above results have been added to the revised manuscript (see **SI: Fig.S22**). Further well-designed studies are warranted to test the role of pentose phosphate pathway of the human gut microbiome in SARS-CoV-2 infection.

Pathway abundances quantified by HUMAnN3 are automatically stratified into contributions from known and uncharacterized microbes at the **species** level. To explore which **strain** has the potential to use the pentose phosphate pathway, we annotated all the nrMAGs using Prokka²³ (a command line software tool to fully annotate a draft bacterial genome in about 10 min on a typical desktop computer) and MicrobeAnnotator²⁴ (a fully automated, easy-to-use pipeline for the comprehensive functional annotation of microbial genomes that combines results from several reference protein databases). Those microbial genomes with high KEGG module completeness of pentose phosphate pathway related modules, i.e., module M00004 (Pentose phosphate pathway) and module M00006 (Pentose phosphate pathway, oxidative phase), represent strains that have the potential to use this pathway.

For module M00004 (**Fig.R9**), among all the nrMAGs, we identified 46 of them that have the highest module completeness (which is 87.5%), including multiple strains from *Bacteroides uniformis* (n=6), *Bacteroides thetaiotaomicron* (n=5), *Bacteroides xylanisolvens* (n=4), *Bacteroides ovatus* (n=3), *Bacteroides salyersiae* (n=2), *Enterobacter himalayensis* (n=2), *Enterobacter roggenkampii* (n=2), *Enterococcus faecalis* (n=2), and *Escherichia coli* (n=2). For module M00006 (**Fig.R10**), we identified 177 nrMAGs that have 100% module completeness, including multiple strains from *Bacteroides uniformis* (n=12), *Bacteroides thetaiotaomicron* (n=11), *Bacteroides ovatus* (n=10), *Streptococcus parasanguinis_B* (n=9), *Bacteroides xylanisolvens* (n=8), *Enterocloster bolteae* (n=7), *Bacteroides fragilis_A* (n=4), *Eisenbergiella massiliensis* (n=4), *Enterocloster aldenensis* (n=4), and *Streptococcus sp000479315* (n=4). Interestingly, 45 of the 46 nrMAGs with 87.5% module completeness on M00004 also showed high module completeness (100%) on module M00006.

Those findings have been added to the revised manuscript (**see SI: FigS.20-22**).

Minor concerns:

1. Figure 3: The sample size of each group (each bar) needs to be clarified in the figure or figure legend.

We thank Reviewer #2 for this excellent suggestion. We have indicated the sample size of each group in Figure3 (see **main text: page 28, Fig.3**).

2. Figure 5: The color bar indicating the Spearman correlation coefficient is unable to be found in Figure 5b.

We thank Reviewer #2 for this comment. In the previous Fig. 5b (which is now Fig.6b in the revised manuscript), the colors of the taxonomical label represent permissive (red), protective (blue), or neutral (gray) nrMAGs. We have now added a color bar to indicate the Spearman correlation coefficient (see **main text: page 31, Fig.6**).

Finally, we would like to Reviewer #2 again for reviewing our manuscript and highly appreciate those very insightful and constructive comments, which have helped us significantly improve the quality of our manuscript. We hope our responses above have addressed all the comments in a satisfactory manner.

Response to Reviewer #3

This work describes the analysis of whole-metagenome shotgun sequencing data with reconstruction of metagenome-assembled genomes (MAGs) from COVID and non-COVID patients from several Chinese cohorts. The study includes both fecal and nasopharyngeal samples.

We thank Reviewer #3 for reviewing our manuscript. We next address each of the reviewer's comments in order.

The authors focus most of their analyses on 1-2 cohorts that include control individuals and therefore the power of the study is reduced compared to what initially the reader would think.

We thank Reviewer #3 for this critical comment.

In the previous manuscript, we collected the raw WMS sequencing data of 514 microbiome samples from 6 publicly available cohorts. The reasons why we mainly focused on two cohorts (Zuo et al. and Yeoh et al.) are two-fold: (1) their nice case-control study design; (2) their relatively large sample sizes. Although the microbiome data from the other 4 cohorts were not included in the majority of our statistical analyses, they still served as key resources for the construction of the COVID-19 related microbial genome catalog.

In the revised manuscript, we analyzed microbiome data from three additional case-control cohorts: Zhang et al. (Total: 207, COVID-19: n=129, Non-COVID-19: n=78), Xu et al. (Total: 69, COVID-19: n=38, Non-COVID-19: n=31), and Li et al. (Total: 65, COVID-19: n=46, Non-COVID-19: n=19) as validation cohorts. We think this has significantly enhanced the power of our study.

Several concerns arise from reading this manuscript.

First of all, the attempt to have microbiome signatures from COVID and non-COVID types is not convincing. In fact, several species associate with one or the other phenotype but such species vary depending on the cohort, the results between datasets are not consistent.

We thank Reviewer #3 for raising this important point.

We agree with the reviewer that the important microbiome features to distinguish COVID-19 from Non-COVID-19 controls are quite different (10% overlapped nrMAGs among the top 30 important features) between two discovery cohorts (Zuo et al. and Yeoh et al.). The inconsistency of important features between the two datasets may be due to the way we split the data, the different sample sizes, and the inherent heterogeneity between different cohorts.

To further address this issue, we performed cross-validation between the two discovery cohorts. Briefly, we trained our machine learning model with samples from the cohort of Zuo et al. (or Yeoh et al.) and then test the model with samples from the cohort of Yeoh et al. (or Zuo et al.),

respectively. The machine learning model trained with the data of Zuo et al. achieved an overall classification performance with AUROC~0.798 and AUPRC~0.607 when tested with samples from the study of Yeoh et al. (**Fig.R11a**). Remarkably, we found the machine learning model trained with samples from Yeoh et al. can almost perfectly distinguish COVID-19 patients from Non-COVID-19 controls in the study of Zuo et al (AUROC~0.9994; AUPRC~0.9991, **Fig.R11b**).

To better understand those results, we also outputted the top-30 most important features ranked based on their mean decrease accuracy (MDA). Here, the MDA of a feature means its average accuracy loss after excluding this feature from the model. We found that the most important nrMAGs identified from the study of Zuo et al. have quite distinct abundance distributions between cases and controls in the study of Zuo et al., but not in the study of Yeoh et al (**Fig.R12a,b**). This explains the lower performance of the model (trained with data from Zuo et al.) in testing data from Yeoh et al. Moreover, the most important nrMAGs identified from the study of Yeoh et al. showed distinct abundance distributions between cases and controls in the study Zuo et al. (**Fig.R12c,d**). This explains the almost perfect performance of the model (trained with data from Yeoh et al.) in testing data from Zuo et al. The above results have been added to the revised manuscript (see **SI: Fig.S14** and **Fig.S15**).

The above results are consistent with the fact that the sample size of training data is a dominant factor affecting a machine learning model's performance in practice²⁵. In principle, the models trained with a larger sample size is more reliable. In this spirit, we also tested the performance of the models (trained with samples from Yeoh et al.) in testing samples from three additional validation cohorts (i.e., Zhang et al., Xu et al., and Li et al.). We found that the machine learning model trained with the samples of Yeoh et al. achieved an overall classification performance with AUROC~0.979 and AUPRC~0.976 in the cohort of Zhang et al, respectively (**Fig.R7a**); AUROC~0.660 and AUPRC~0.571 in the cohort of Xu et al. (**Fig.R7b**); AUROC~0.699 and AUPRC~0.565 in the cohort of Li et al. (**Fig.R7c**). Therefore, the microbiome features of COVID-19 identified from our study are relatively generic. The above results have been added to the revised manuscript (see **SI: Fig.S16**).

The diversity difference is only explored at indexes level. Since the authors also have MAGs that are resolved at strain level, it could be interesting to understand whether there's strain-level diversity within the same species according to covid vs non-covid phenotypes (or even with severity stratification). In other words, how many strains of the same species are found in covid and non-covid cases? Is there a strain dominance in species associated to the covid subjects?

We thank Reviewer #3 for this very insightful comment.

To explore the idea of strain-level diversity within the same species, we analyzed the data from the two discovery cohorts (Zuo et al. and Yeoh et al.), as well as the three validation cohorts (Zhang et al., Xu et al., and Li et al.). We first grouped all the nrMAGs to the species level based on the GTDB taxonomy annotation. For each species, we computed its *strain richness* (i.e., the number its nrMAGs) for all microbiome samples. Those nrMAGs without the species annotation

and species containing only one nrMAG were excluded in downstream analyses. Interestingly, the top-30 microbial species with the highest strain richness were highly overlapped in the discovery and validation cohorts (**Figs.R1-R3**).

Notably, we found, for the first time, that COVID-19 patients lost many strains for certain microbial species when compared to Non-COVID-19 controls in both the discovery (**Fig.R1**) and validation cohorts (**Fig.R3**). This finding suggests that SARS-CoV-2 infection is associated with decreasing strain richness of certain species. Note that those species with noticeable change of strain richness are highly overlapped between the two discovery cohorts (**Fig.R4a**), including multiple protective microbial species identified by our GMPT pipeline, such as *Bariatricus comes*, *Blautia_A obeum*, *Blautia_A wexlerae*, *Dorea formicigenerans*, *Faecalibacterium prausnitzii_D*, *Faecalibacterium sp900539945*, and *Fusicatenibacter saccharivorans*. Importantly, these results are highly consistent with the alpha diversity analysis at the nrMAGs level that COVID-19 patients had significantly lower number of nrMAGs than that of Non-COVID-19 controls in the discovery cohorts. We found that some microbial species (9 of 30) with high COVID-19 related change of strain richness in the discovery cohorts were also identified in the validation cohorts (**Fig.R4c**). Note that, for the Xu et al. cohort, several species (e.g., *Anaerobutyricum hallii*, *Bacteroides stercoris*, and *Blautia_A obeum*) showed significantly high strain richness in COVID-19 patients than in Non-COVID-19 controls (**Fig.R3e**). This may be due to the fact that microbiome samples of COVID-19 patients (collected in 2020) and Non-COVID-19 control (collected in 2016) were not collected and sequenced at the same time. The above results have been added to the revised manuscript (see **main text: Fig.4 and FigS10-12**).

We next investigated the disease severity in relation to the strain richness using data from Yeoh et al. We found 7 microbial species whose strain richness were positively correlated with disease severity (Spearman correlation coefficients ≥ 0.9). Of these 7 species, 2 (i.e., *Enterocloster bolteae* and *Hungatella effluvii*) have been identified by our GMPT pipeline as species to which permissive nrMAGs belongs. Moreover, a total of 222 microbial species' strain richness were negatively correlated with disease severity (Spearman correlation coefficients ≤ -0.9), and these species covered almost all species (17 of 21) identified by our GMPT pipeline to which protective nrMAGs belongs, including *Blautia_A obeum*, *Bariatricus comes*, *Blautia_A wexlerae*, and *Faecalibacterium prausnitzii_D*. The above results have been added to the revised manuscript (see **SI: Table S2**).

Regarding whether there is a strain dominance of microbial species in the gut microbiome associated with COVID-19 patients, we performed analyses in two different scenarios and summarized our results as follows.

- Scenario-1: focusing on species identified from the microbiome samples of COVID-19 patients. In this scenario, we first excluded those species containing only one nrMAG (strain), because in this case we cannot determine if this is due to that (a) this species contains only one strain; or (b) other strains have been outcompeted by the surviving one. For each of the remaining species, we counted its actual number of strains n_{actual} , and calculated its *effective number of strains* $n_{\text{effective}}$ (i.e., the number of equally-common strains required to give a particular value of an alpha-diversity index, e.g., the Shannon

index, for the strain abundance profile of this species)²⁶ for COVID-19 patients in the two discovery cohorts and the three validation cohorts. For each cohort, we found many species with very large ($n_{\text{actual}} - n_{\text{effective}}$) values, suggesting the presence of dominating strains for those species (**Fig.R13**).

- Scenario-2: focusing on differentially abundant species identified from the comparison between COVID-19 patients and Non-COVID-19 controls. In this scenario, we first grouped all nrMAGs to species level based on the GTDB taxonomy annotation. Differential abundance analyses were conducted between COVID-19 patients and Non-COVID-19 controls in the two discovery cohorts and the three validation cohorts using ANCOM²⁷. A total of 92, 126, 81, 129, and 20 species were identified as differentially abundant species in the studies of Zuo et al., Yeoh et al., Zhang et al., Xu et al., and Li et al, respectively. We then excluded those species containing only one nrMAG (strain). For each of the remaining differentially abundant species, we counted its actual number of strains n_{actual} , and calculated its *effective number of strains* $n_{\text{effective}}$ in samples from COVID-19 patients. Similar to scenario-(1), we found that many species with very large ($n_{\text{actual}} - n_{\text{effective}}$) values, suggesting the presence of dominating strains for those species (**Fig.R14**).

Finally, the functional metagenomics part is potentially interesting but not enough exploited or interpreted.

We thank Reviewer #3 for this critical comment. Reviewer #2 also raised a similar comment. To avoid repetitive response, please see **pages 7-8** of this letter for our detailed response.

The discussion lacks of proper results' interpretation and is in some paragraphs too repetitive of the results section.

We thank Reviewer #3 for this comment.

We have heavily revised the Discussion section to make it complementary to (rather than repetitive of) the Results section. In particular, we discuss the cross and external validation of microbiome signatures of COVID-19 and potential links between pentose phosphate pathway of microbial community and SARS-CoV-2 infection (see **main text: page 11, lines: 424-432; pages 12-13, lines 447-451 and 481-509**).

The study appears overall preliminary and the proposed microbiome signature of COVID disease and progression are not convincing.

We thank Reviewer #3 for this critical comment.

In this study, we presented the construction of COVID-19 related microbial genome catalog using assembly and reference free binning of metagenome for the first time. Given the advanced strain

information and the ability to identify new species, we believe the application of this strategy is highly novel and the data presented is important to the field. Following the reviewer's previous comment, we performed the cross-validation between the data sets of Zuo et al. and Yeoh et al. and found that COVID-19 related microbiome features identified from a larger cohort (i.e. Yeoh et al.) can be validated in a smaller cohort (i.e., Zuo et al.) (**Fig.R11**).

Moreover, we also tested the generalization of COVID-19 microbiome features on three validation cohorts (i.e., Zhang et al., Xu et al., and Li et al.). We found that the machine learning model trained with the data of Yeoh et al. achieved an overall reasonable classification performance on three external validation datasets (**Fig.R7**). Therefore, these findings demonstrate that the microbiome features identified in this study might offer relatively universal utility as a diagnostic test for COVID-19.

Using the microbiome data and machine learning model, we were able to predict the date of negative RT-qPCR result of patients with COVID-19 in the study of Yeoh et al. This analysis indicated that several microbial species (species level annotation of nrMAGs) may interact with the progression of COVID-19 (see **main text: Fig.5b**) that have been previously reported involved in COVID-19 including *Citrobacter freundii*²⁸, *Veillonella parvula*^{12,29,30}, and *Parabacteroides distasonis*^{9,12}. Therefore, our prediction of COVID-19 progression is partially supported by the existing literature evidence. Notably, we observed some opportunistic pathogens were associated with the progression of COVID-19, including nrMAGs from *Klebsiella quasivariicola*³¹, *Klebsiella pneumoniae*³², and *Escherichia coli*³³, supporting the possibility that secondary infections by opportunistic pathogens may affect the progression of COVID-19³⁴⁻³⁷.

However, due to the lack of microbiome samples collected before and after negative RT-QPCR result or exact date information of that in other discovery and validation cohorts, we were unable to validate the generalization of the microbiome feature of COVID-19 progression. Further studies are needed to validate these findings and determine how those microbes influence the progression of COVID-19. We have now added the following sentence to the revised manuscript (see **main text: page 12, lines 463-464**).

“However, further studies are needed to validate these findings and determine how those microbes influence the progression of COVID-19.”

Specific comments:

168-174. Please consider erasing this section as it is redundant compared to fig. 2d. Moreover, these are the common phyla human microbes belong to.

We thank Reviewer #3 for this comment. We have revised this part accordingly (see **main text: page 5, lines 175-176**).

176-192. Looking at microbial diversity in COVID and non COVID patients is surely interesting.

However, the results show no clear trend of differences. The choice of separating the studies is arguable, the reader would think that considering all subjects together there are no differences?

We thank Reviewer #3 for this comment.

In the previous manuscript, we demonstrated that the alpha diversity (i.e., richness and Shannon index) analysis of the gut microbiome in COVID-19 patients were significantly lower than that in Non-COVID-19 health controls in two datasets (i.e., Zuo et al. and Yeoh et al.). Although we lost this signal in the nasopharyngeal microbiome, this may be due to the small sample size, unhealthy controls, and sequencing quality.

Following the reviewer's suggestion, we pooled samples from different datasets in both discovery cohorts (6 datasets: **Fig.R15a,b**) and validation cohorts (3 datasets: **Fig.R15c,d**). Interestingly, pooling samples did not affect the alpha diversity comparison between COVID-19 and Non-COVID-19 in both discovery and validation cohorts. We still found that the alpha diversity of the gut microbiome in COVID-19 patients were significantly lower than that in Non-COVID-19 health controls.

184. This is not true for the Zuo cohort.

We thank Reviewer #3 for this scrupulous check. We apologize for the mislabeling of Non-COVID-19 controls and Pneumonia cases in Fig.3a and Fig.3b (Zuo et al.). We have corrected Fig.3 accordingly now (see **main text: page 28, Fig.3**).

194-199. I still don't see a clear difference here. It appears that non-covid patients are more similar while there's an increased variability in covid individuals, this is the result I see here. In addition, Fig.3d,e,g,h could be left out.

We thank Reviewer #3 for this insightful comment. Ordination techniques, such as principal coordinates analysis (PCoA), reduce the dimensionality of microbiome data sets so that a summary of the beta diversity relationships can be visualized in two- or three-dimensional scatterplots. Observations based on PCoA plots should be substantiated with statistical analyses that assess the clusters. Here, we applied the Permutational Multivariate Analysis of Variance (PERMANOVA) with 9999 permutations and confirmed that clustering patterns between COVID-19 and Non-COVID-19 were statistically significant (P value <0.05).

In the revised manuscript, we moved Fig.3d,e,g,h to SI (see **SI: Fig.S8**).

223-245. The fact that the COVID signature species are different between datasets makes the COVID signature concept and the possibility to have COVID microbiome markers quite weak.

We thank Reviewer #3 for this critical comment. As we responded to the previous comment (please see pages 10-11 of this response letter), we addressed this issue with further cross-validation on two discovery cohorts (Zuo et al. and Yeoh et al) and three validation cohorts (Zhang et al., Xu et al., and Li et al.). We found that the classification models trained with the data of Yeoh et al. (which has the largest sample size) achieved an overall reasonable classification performance on one discovery cohort (Zuo et al., **Fig.R11**) and three external validation datasets (**Fig.R7**). Therefore, the identified microbiome features of COVID-19 are relatively generic in our study.

298-320. Are these also taken from 1 dataset only? In addition, also in the case of functional potential, the associations with permissive and protective cases seems quite random and not well explained.

We thank Reviewer #3 for this important comment. We apologize for not clearly explaining the logic here. In the previous manuscript, for the genome annotation of nrMAGs, we only focused on the data from Yeoh et al. This is because that is the only cohort that has the disease severity data. Through the GMPT pipeline, we identified a set of putative permissive and protective nrMAGs. A natural question that arises is how those permissive and protective nrMAGs may interact with SARS-CoV-2 infection? We hypothesized that those nrMAGs may interact with COVID-19 through some specific functional pathways. Therefore, we annotated those nrMAGs and found that permissive nrMAGs showed significantly higher completeness level at the pentose phosphate pathway compared to protective nrMAGs. We have now added the following sentence to the revised manuscript (see **main text: page 10, lines 369-370**).

“To understand how those permissive and protective nrMAGs identified from the study of Yeoh et al.¹⁹ may interact with SARS-CoV-2 infection, we next investigated whether the functional capacity of permissive and protective nrMAGs differ.”

399-405. These results do not indicate a clear set of metabolic pathways that associate with a permissive or protective signature and that can explain the development of the covid disease. There is no clear idea on how the gut microbes can interact with the virus infection or how they can change their metabolic potential with progression of the virus.

We thank Reviewer #3 for this critical comment.

Regarding metabolic pathways, in addition to the genome annotation of permissive and protective nrMAGs in our previous manuscript, here we applied HUMAnN3²² to perform functional profiling at the community level. Importantly, we found that the abundance of pentose phosphate pathway (PENTOSE-P-PWY) in COVID-19 patients was significantly higher than that in the Non-COVID-19 controls in both studies of Zuo et al. (**Fig.R8a**) and Yeoh et al. (**Fig.R8b**). Among three validation cohorts, we found that the pentose phosphate pathway showed higher abundance in COVID-19 patients in the study of Zhang et al. (p value=0.39, **Fig.R8c**) and Xu et al. (p value=0.013, **Fig.R8d**). Overall, these results suggest that the pentose phosphate pathway may

play some role in the SARS-CoV-2 infection. Importantly, a previous study reported a significant increase in the levels of some intermediates of the glycolytic and pentose phosphate pathways in sera of COVID-19 positive patients³⁸. Moreover, SARS-CoV-2 infection was found to be associated with changes in the regulation of the pentose phosphate pathway in both *in vivo* (Caco-2 cells)³⁹ and *in vitro* (ferret model)⁴⁰ studies.

The pentose phosphate pathway is an important physiological process that can occur in 2 phases: oxidative and nonoxidative. Reactions of the pentose phosphate pathway, occur virtually ubiquitously, and maintain a central metabolic role in providing the RNA backbone precursors ribose 5-phosphate and erythrose 4-phosphate as precursors for aromatic amino acids⁴¹. The aromatic amino acids in the juxtamembrane domain of the SARS-CoV S glycoprotein play critical roles in receptor-dependent virus-cell and cell-cell fusion⁴². A previous study reported that the UK mutation (N501Y: a mutation from asparagine to tyrosine conferring one more aromatic amino acid to receptor binding domain) interacts closely with Y41 (ACE2) in the receptor therefore producing aromatic-aromatic interactions that provide for stronger binding between receptor and spike⁴³. Indeed, the levels of aromatic amino acids (e.g., tyrosine, phenylalanine, and tryptophan) were increased significantly in COVID-19 patients compared with controls using targeted metabolic analysis⁴⁴. Overall, these results suggest that specific microbes (permissive nrMAGs, such as strains from *Hungatella effluvii* and *Enterocloster bolteae*) may play a role in mediating SRAS-CoV-2 entry into host cells through pentose phosphate pathway and aromatic amino acids. However, further mechanistic studies are warranted to test the exact role of our candidate permissive and protective nrMAGs in SARS-CoV-2 infection.

We have now added the above discussion to the Discussion section of the revised manuscript (see main text: page 13, lines 481-509).

“In addition to the genome annotation of permissive and protective nrMAGs, we also found that the overall abundance of pentose phosphate pathway in COVID-19 patients was higher than that in the Non-COVID-19 controls in two discovery cohorts and two validation cohorts. The pentose phosphate pathway is an important physiological process that can occur in 2 phases: oxidative and nonoxidative. Reactions of the pentose phosphate pathway, occur virtually ubiquitously, and maintain a central metabolic role in providing the RNA backbone precursors ribose 5-phosphate and erythrose 4-phosphate as precursors for aromatic amino acids⁶⁸. The aromatic amino acids in the juxtamembrane domain of the SARS-CoV S glycoprotein play critical roles in receptor-dependent virus-cell and cell-cell fusion⁶⁹. A previous study reported that the UK mutation (N501Y: a mutation from asparagine to tyrosine conferring one more aromatic amino acid to receptor binding domain) interacts closely with Y41 (ACE2) in the receptor therefore producing aromatic-aromatic interactions that provide for stronger binding between receptor and spike⁷⁰. Indeed, the levels of aromatic amino acids (e.g., tyrosine, phenylalanine, and tryptophan) were increased significantly in COVID-19 patients compared with controls using targeted metabolic analysis⁷¹.

Importantly, a previous study reported a significant increase in the levels of some intermediates of the glycolytic and pentose phosphate pathways in sera of COVID-19 positive patients⁶⁵. Moreover, an earlier study (86 COVID-19 patients and 57 healthy controls, United Arab Emirates) reported that the pentose phosphate pathway was significantly upregulated on COVID-19 patient microbiome samples using 16S rRNA gene sequencing together with a phylogenetic investigation of communities by reconstructing unobserved state (PICRUSt)⁷². In addition, SARS-CoV-2 infection was found to be associated with changes in the regulation of the pentose phosphate pathway in both in vivo (Caco-2 cells)⁶⁶ and in vitro (ferret model)⁶⁷ studies. Together, these results suggest that specific microbes (permissive nrMAGs, such as strains from *Hungatella effluvii* and *Enterocloster bolteae*) may play a role in mediating SRAS-CoV-2 entry into host cells through pentose phosphate pathway and aromatic amino acids. However, further mechanistic studies are warranted to test the exact role of our candidate permissive and protective nrMAGs in SARS-CoV-2 infection.”

Finally, we would like to Reviewer #3 again for reviewing our manuscript and highly appreciate those very insightful and constructive comments, which have helped us significantly improve the quality of our manuscript. We hope our responses above have addressed all the comments in a satisfactory manner.

Response figure and legend

Figure R1. COVID-19 related changes in strain richness of microbial species in the two discovery cohorts. a,b: The top-30 species with the highest strain-richness (i.e., number of nrMAGs) identified from the study of Zuo et al. (a) and Yeoh et al. (b). c,d: The top-30 species with the highest strain-richness change between the Non-COVID-19 (blue) and COVID-19 (red) samples identified from the study of Zuo et al. (c) and Yeoh et al. (d). Data are presented as mean \pm standard error of mean. *P* values were calculated by two-sided Wilcoxon–Mann–Whitney test.

Figure R2. Venn diagram of top-30 species with the highest strain richness identified from different cohorts. **a**, two discovery cohorts. **b**, three validation cohorts. **c**, two discovery and three validation cohorts.

Figure R3. COVID-19 related changes in strain richness of microbial species in the three validation cohorts. **a-c:** The top-30 species with the highest strain richness (i.e., the number of nrMAGs) identified from the study of Zhang et al. (a), Xu et al. (b), and Li et al. (c). **d-f:** The top-30 species with the highest strain richness change between the Non-COVID-19 (blue) and COVID-19 (red) samples identified from the study of Zhang et al. (d), Xu et al. (e), and Li et al. (f). Data are presented as mean \pm standard error of mean. *P* values were calculated by two-sided Wilcoxon–Mann–Whitney test.

Figure R4. Venn diagram of top-30 species with the highest COVID-19 related strain-richness change identified from different cohorts. **a**, two discovery cohorts. **b**, three validation cohorts. **c**, two discovery and three validation cohorts.

Figure R5. Alpha diversity analyses of COVID-19 related human microbiome samples in the three validation cohorts. a-c: Richness of the human microbiome at the nrMAG-level from the study of Zhang et al. (a), Xu et al. (b), and Li et al. (c). **d-f:** Shannon index of the human microbiome at the nrMAG-level from the study of Zhang et al. (d), Xu et al. (e), and Li et al. (f). *P* values were calculated by two-sided Wilcoxon–Mann–Whitney test.

Figure R6. Beta diversity analyses of COVID-19 related human microbiome samples in the three validation cohorts. a-c: Principal Coordinates Analysis (PCoA) plot based on Bray–Curtis dissimilarity of microbial compositions from the study of Zhang et al. (a), Xu et al. (b), and Li et al. (c). All PERMANOVA tests were performed with 9999 permutations based on Bray–Curtis dissimilarity. **d-f:** Within-group Bray-Curtis dissimilarity of the human microbiome at the nrMAG-level from the study of Zhang et al. (d), Xu et al. (e), and Li et al. (f). P values were calculated by two-sided Wilcoxon–Mann–Whitney test.

Figure R7. External validation of the machine learning model. The machine learning model was trained on the data from Yeoh et al. and then tested on the data from Zhang et al. (a), Xu et al. (b), and Li et al. (c). The bars represent mean \pm standard deviation of two standard classification performance metrics: the Area Under the Receiver Operating Characteristic (AUROC) curve, and the Area Under the Precision-Recall Curve (AUPRC).

Figure R8. Abundance comparison of the pentose phosphate pathway between COVID-19 patients and Non-COVID-19 controls. The cohort from the study of Zuo et al. (a), Yeoh et al. (b), Zhang et al. (c), Xu et al. (d), and Li et al. (e). *P* values were calculated by two-sided Wilcoxon–Mann–Whitney test.

Figure R9. The phylogenetic tree of strains (nrMAGs) that have the potential to use the pentose phosphate pathway (Pentose phosphate cycle, M00004, module completeness: 87.5%). The phylogenetic tree of nrMAGs was constructed using PhyloPhlAn⁴⁵ and visualized using iTOL⁴⁶. The color of cycle and clades represents phylum.

Figure R10. The phylogenetic tree of strains (nrMAGs) that have the potential to use the pentose phosphate pathway (oxidative phase, glucose 6P => ribulose 5P, M00006, module completeness: 100%). The phylogenetic tree of nrMAGs was constructed using PhyloPhlAn⁴⁵ and visualized using iTOL⁴⁶. The color of cycle and clades represents phylum.

Figure R11. Cross-validation of the machine learning model between the two discovery cohorts. **a**, The machine learning model was trained on the data from Zuo et al. and then tested on the data from Yeoh et al. **b**, The machine learning model was trained on the data from Yeoh et al. and then tested on the data from Zuo et al. The bars represent mean \pm standard deviation of two standard classification performance metrics: the Area Under the Receiver Operating Characteristic (AUROC) curve, and the Area Under the Precision-Recall Curve (AUPRC).

Figure R12. Heat map of top-30 most important nrMAGs related to the performance of cross-validation. a,b: The relative abundances of the most important features in the machine learning model (trained on the data from Zuo et al.) in the microbiome samples of Zuo et al. (a) and Yeoh et al. (b). c,d: The relative abundances of the most important features in the machine learning model (trained on the data from Yeoh et al.) in the microbiome samples of Zuo et al. (c) and Yeoh et al. (d). The importance of each feature was quantified by the Mean Decrease in Accuracy (MDA) of the classifier due to the exclusion (or permutation) of this feature.

Figure R13. Representative microbial species with strong strain dominance in COVID-19 patients. Different cohorts: Zuo et al. (a), Yeoh et al. (b), Zhang et al. (c), Xu et al. (d), and Li et al. (e). For each species, we demonstrate its actual number of strains n_{actual} (dark blue) and its effective number of strains $n_{\text{effective}}$ (light blue). Bars represent mean \pm standard error of mean. The number following the name of each species indicates the number of samples that have at least one nrMAG (strain) of this species. For each cohort, we show the top-30 species in descending order of their ($n_{\text{actual}} - n_{\text{effective}}$) values. Note that only species present in at least 20% samples were included.

Figure R14. Representative differential abundant microbial species with strong strain dominance in COVID-19 patients. Different cohorts: Zuo et al. (a), Yeoh et al. (b), Zhang et al. (c), Xu et al. (d), and Li et al. (e). For each differential abundant species (identified using ANCOM²⁷), we demonstrate its actual number of strains n_{actual} (dark blue) and its effective number of strains $n_{\text{effective}}$ (light blue). Bars represent mean \pm standard error of mean. The number following each species name indicates the number of samples that have at least one nrMAG (strain) of this species. For each cohort, we show species (up to 30) in descending order of their ($n_{\text{actual}} - n_{\text{effective}}$) values. Note that only species present in at least 20% samples were included. For the cohorts of Zuo et al. and Li et al., we only identified 17 and 11 microbial species present in at least 20 % samples, respectively.

Figure R15. Comparison of alpha diversity on the pooled COVID-19 subjects and Non-COVID-19 controls on discovery cohorts (a-b) and validation cohorts (c-d). Richness (a,c). Shannon (b,d). P-values were calculated by two-sided Wilcoxon–Mann–Whitney test.

References

- 1 Koo, H. & Morrow, C. D. Early indicators of microbial strain dysbiosis in the human gastrointestinal microbial community of certain healthy humans and hospitalized COVID-19 patients. *Sci Rep* **12**, 6562, doi:10.1038/s41598-022-10472-w (2022).
- 2 Hoque, M. N. *et al.* Diversity and genomic determinants of the microbiomes associated with COVID-19 and non-COVID respiratory diseases. *Gene Rep* **23**, 101200, doi:10.1016/j.genrep.2021.101200 (2021).
- 3 Hong, C. *et al.* PathoScope 2.0: a complete computational framework for strain identification in environmental or clinical sequencing samples. *Microbiome* **2**, 33, doi:10.1186/2049-2618-2-33 (2014).
- 4 Surana, N. K. & Kasper, D. L. Moving beyond microbiome-wide associations to causal microbe identification. *Nature* **552**, 244-247, doi:10.1038/nature25019 (2017).
- 5 Ke, S. *et al.* A Computational Method to Dissect Colonization Resistance of the Gut Microbiota against Pathogens. *bioRxiv* (2022).
- 6 Wu, Y. *et al.* Altered oral and gut microbiota and its association with SARS-CoV-2 viral load in COVID-19 patients during hospitalization. *NPJ Biofilms Microbiomes* **7**, 61, doi:10.1038/s41522-021-00232-5 (2021).
- 7 Zuo, T. *et al.* Alterations in Gut Microbiota of Patients With COVID-19 During Time of Hospitalization. *Gastroenterology* **159**, 944-955 e948, doi:10.1053/j.gastro.2020.05.048 (2020).
- 8 Liu, Q. *et al.* Gut microbiota dynamics in a prospective cohort of patients with post-acute COVID-19 syndrome. *Gut* **71**, 544-552, doi:10.1136/gutjnl-2021-325989 (2022).
- 9 Khan, M. *et al.* Gut Dysbiosis and IL-21 Response in Patients with Severe COVID-19. *Microorganisms* **9**, doi:10.3390/microorganisms9061292 (2021).
- 10 Tang, L. *et al.* Clinical Significance of the Correlation between Changes in the Major Intestinal Bacteria Species and COVID-19 Severity. *Engineering (Beijing)* **6**, 1178-1184, doi:10.1016/j.eng.2020.05.013 (2020).
- 11 Wu, C. *et al.* The volatile and heterogeneous gut microbiota shifts of COVID-19 patients over the course of a probiotics-assisted therapy. *Clin Transl Med* **11**, e643, doi:10.1002/ctm2.643 (2021).
- 12 Yeoh, Y. K. *et al.* Gut microbiota composition reflects disease severity and dysfunctional immune responses in patients with COVID-19. *Gut* **70**, 698-706, doi:10.1136/gutjnl-2020-323020 (2021).
- 13 Hazan, S. *et al.* Lost microbes of COVID-19: Bifidobacterium, Faecalibacterium depletion and decreased microbiome diversity associated with SARS-CoV-2 infection severity. *BMJ Open Gastroenterol* **9**, doi:10.1136/bmjgast-2022-000871 (2022).
- 14 Li, S. *et al.* Microbiome Profiling Using Shotgun Metagenomic Sequencing Identified Unique Microorganisms in COVID-19 Patients With Altered Gut Microbiota. *Front Microbiol* **12**, 712081, doi:10.3389/fmicb.2021.712081 (2021).
- 15 Sun, Z. *et al.* Gut microbiome alterations and gut barrier dysfunction are associated with host immune homeostasis in COVID-19 patients. *BMC Med* **20**, 24, doi:10.1186/s12916-021-02212-0 (2022).
- 16 Venzon, M. *et al.* Gut microbiome dysbiosis during COVID-19 is associated with increased risk for bacteremia and microbial translocation. *bioRxiv*, 2021.2007.2015.452246, doi:10.1101/2021.07.15.452246 (2022).

- 17 Xu, X. *et al.* Integrated analysis of gut microbiome and host immune responses in COVID-19. *Front Med* **16**, 263-275, doi:10.1007/s11684-022-0921-6 (2022).
- 18 Zhang, F. *et al.* Prolonged Impairment of Short-Chain Fatty Acid and L-Isoleucine Biosynthesis in Gut Microbiome in Patients With COVID-19. *Gastroenterology*, doi:10.1053/j.gastro.2021.10.013 (2021).
- 19 Bai, X. *et al.* Characterization of the Upper Respiratory Bacterial Microbiome in Critically Ill COVID-19 Patients. *Biomedicines* **10**, doi:10.3390/biomedicines10050982 (2022).
- 20 Sulaiman, I. *et al.* Microbial signatures in the lower airways of mechanically ventilated COVID-19 patients associated with poor clinical outcome. *Nat Microbiol* **6**, 1245-1258, doi:10.1038/s41564-021-00961-5 (2021).
- 21 Warmbrunn, M. V., Attaye, I., Herrema, H. & Nieuwdorp, M. Protocol Standardization of Microbiome Studies-Daunting but Necessary. *Gastroenterology* **162**, 1822-1824, doi:10.1053/j.gastro.2022.03.017 (2022).
- 22 Beghini, F. *et al.* Integrating taxonomic, functional, and strain-level profiling of diverse microbial communities with bioBakery 3. *Elife* **10**, doi:10.7554/eLife.65088 (2021).
- 23 Seemann, T. Prokka: rapid prokaryotic genome annotation. *Bioinformatics* **30**, 2068-2069, doi:10.1093/bioinformatics/btu153 (2014).
- 24 Ruiz-Perez, C. A., Conrad, R. E. & Konstantinidis, K. T. MicrobeAnnotator: a user-friendly, comprehensive functional annotation pipeline for microbial genomes. *BMC Bioinformatics* **22**, 11, doi:10.1186/s12859-020-03940-5 (2021).
- 25 Figueroa, R. L., Zeng-Treitler, Q., Kandula, S. & Ngo, L. H. Predicting sample size required for classification performance. *BMC Med Inform Decis Mak* **12**, 8, doi:10.1186/1472-6947-12-8 (2012).
- 26 Cao, Y., Hawkins, C. P. & Magrath, A. Weighting effective number of species measures by abundance weakens detection of diversity responses. *Journal of Applied Ecology* **56**, 1200-1209, doi:10.1111/1365-2664.13345 (2019).
- 27 Mandal, S. *et al.* Analysis of composition of microbiomes: a novel method for studying microbial composition. *Microb Ecol Health Dis* **26**, 27663, doi:10.3402/mehd.v26.27663 (2015).
- 28 Zhou, Y. *et al.* Gut Microbiota Dysbiosis Correlates with Abnormal Immune Response in Moderate COVID-19 Patients with Fever. *J Inflamm Res* **14**, 2619-2631, doi:10.2147/JIR.S311518 (2021).
- 29 Ma, S. *et al.* Metagenomic analysis reveals oropharyngeal microbiota alterations in patients with COVID-19. *Signal Transduct Target Ther* **6**, 191, doi:10.1038/s41392-021-00614-3 (2021).
- 30 Cao, J. *et al.* Integrated gut virome and bacteriome dynamics in COVID-19 patients. *Gut Microbes* **13**, 1-21, doi:10.1080/19490976.2021.1887722 (2021).
- 31 Rodriguez-Medina, N., Barrios-Camacho, H., Duran-Bedolla, J. & Garza-Ramos, U. *Klebsiella variicola*: an emerging pathogen in humans. *Emerg Microbes Infect* **8**, 973-988, doi:10.1080/22221751.2019.1634981 (2019).
- 32 Effah, C. Y., Sun, T., Liu, S. & Wu, Y. *Klebsiella pneumoniae*: an increasing threat to public health. *Ann Clin Microbiol Antimicrob* **19**, 1, doi:10.1186/s12941-019-0343-8 (2020).
- 33 Kaper, J. B., Nataro, J. P. & Mobley, H. L. Pathogenic *Escherichia coli*. *Nat Rev Microbiol* **2**, 123-140, doi:10.1038/nrmicro818 (2004).

- 34 Gupta, A. *et al.* Nasopharyngeal microbiome reveals the prevalence of opportunistic pathogens in SARS-CoV-2 infected individuals and their association with host types. *Microbes Infect* **24**, 104880, doi:10.1016/j.micinf.2021.104880 (2021).
- 35 Xiong, D. *et al.* Enriched Opportunistic Pathogens Revealed by Metagenomic Sequencing Hint Potential Linkages between Pharyngeal Microbiota and COVID-19. *Virol Sin* **36**, 924-933, doi:10.1007/s12250-021-00391-x (2021).
- 36 Chhibber-Goel, J., Gopinathan, S. & Sharma, A. Interplay between severities of COVID-19 and the gut microbiome: implications of bacterial co-infections? *Gut Pathog* **13**, 14, doi:10.1186/s13099-021-00407-7 (2021).
- 37 Yamamoto, S. *et al.* The human microbiome and COVID-19: A systematic review. *PLoS One* **16**, e0253293, doi:10.1371/journal.pone.0253293 (2021).
- 38 Thomas, T. *et al.* COVID-19 infection alters kynurenine and fatty acid metabolism, correlating with IL-6 levels and renal status. *JCI Insight* **5**, doi:10.1172/jci.insight.140327 (2020).
- 39 Bojkova, D. *et al.* Targeting the Pentose Phosphate Pathway for SARS-CoV-2 Therapy. *Metabolites* **11**, doi:10.3390/metabo11100699 (2021).
- 40 Beale, D. J. *et al.* Metabolic Profiling from an Asymptomatic Ferret Model of SARS-CoV-2 Infection. *Metabolites* **11**, doi:10.3390/metabo11050327 (2021).
- 41 Stincone, A. *et al.* The return of metabolism: biochemistry and physiology of the pentose phosphate pathway. *Biol Rev Camb Philos Soc* **90**, 927-963, doi:10.1111/brv.12140 (2015).
- 42 Howard, M. W. *et al.* Aromatic amino acids in the juxtamembrane domain of severe acute respiratory syndrome coronavirus spike glycoprotein are important for receptor-dependent virus entry and cell-cell fusion. *J Virol* **82**, 2883-2894, doi:10.1128/JVI.01805-07 (2008).
- 43 Padilla-Sanchez, V. SARS-CoV-2 Structural Analysis of Receptor Binding Domain New Variants from United Kingdom and South Africa. *Research Ideas and Outcomes* **7**, doi:10.3897/rio.7.e62936 (2021).
- 44 Wu, J., Zhao, M., Li, C., Zhang, Y. & Wang, D. W. The SARS-CoV-2 induced targeted amino acid profiling in patients at hospitalized and convalescent stage. *Biosci Rep* **41**, doi:10.1042/BSR20204201 (2021).
- 45 Asnicar, F. *et al.* Precise phylogenetic analysis of microbial isolates and genomes from metagenomes using PhyloPhlAn 3.0. *Nat Commun* **11**, 2500, doi:10.1038/s41467-020-16366-7 (2020).
- 46 Letunic, I. & Bork, P. Interactive Tree Of Life (iTOL) v5: an online tool for phylogenetic tree display and annotation. *Nucleic Acids Res* **49**, W293-W296, doi:10.1093/nar/gkab301 (2021).

REVIEWERS' COMMENTS

Reviewer #2 (Remarks to the Author):

All my comments have been adequately addressed.

Reviewer #3 (Remarks to the Author):

The authors performed a very careful revision of their manuscript taking thoroughly into account the reviewers comments.